# Effects of 8 Weeks of 2S-Hesperidin Supplementation on Performance in Amateur Cyclists

**DOI:** 10.3390/nu12123911

**Published:** 2020-12-21

**Authors:** Francisco Javier Martínez-Noguera, Cristian Marín-Pagán, Jorge Carlos-Vivas, Pedro E. Alcaraz

**Affiliations:** 1Research Center for High Performance Sport, Campus de los Jerónimos Nº 135, Catholic University of Murcia, UCAM, 30107 Murcia, Spain; cmarin@ucam.edu (C.M.-P.); palcaraz@ucam.edu (P.E.A.); 2Health, Economy, Motricity and Education Research Group (HEME), Faculty of Sport Sciences, University of Extremadura. Avda. de la Universidad, s/n., 10003 Cáceres, Spain; jorge.carlosvivas@gmail.com

**Keywords:** flavonoid, polyphenols, orange extract, performance, endurance, aerobic, anaerobic, nutrigenomic and sport nutrition

## Abstract

2S-Hesperidin is a flavanone (flavonoid) found in high concentrations in citrus fruits. It has an antioxidant and anti-inflammatory effects, improving performance in animals. This study investigated the effects of chronic intake of an orange extract (2S-hesperidin) or placebo on non-oxidative/glycolytic and oxidative metabolism markers and performance markers in amateur cyclists. A double-blind, randomized, placebo-controlled trial was carried out between late September and December 2018. Forty amateur cyclists were randomized into two groups: one taking 500 mg/day 2S-hesperidin and the other taking 500 mg/day placebo (microcellulose) for eight weeks. All participants completed the study. An incremental test was used to evaluate performance, and a step test was used to measure oxygen consumption, carbon dioxide, efficiency and oxidation of carbohydrates and fat by indirect calorimetry. The anaerobic power (non-oxidative) was determined using Wingate tests (30 s). After eight weeks supplementation, there was an increase in the incremental test in estimated functional threshold power (FTP) (3.2%; *p* ≤ 0.05) and maximum power (2.7%; *p* ≤ 0.05) with 2S-hesperdin compared to placebo. In the step test, there was a decrease in VO_2_ (L/min) (−8.3%; *p* ≤ 0.01) and VO_2_R (mL/kg/min) (−8.9%; *p* ≤ 0.01) at VT2 in placebo. However, there were no differences between groups. In the Wingate test, there was a significant increase (*p* ≤ 0.05) in peak and relative power in both groups, but without differences between groups. Supplementation with an orange extract (2S-hesperdin) 500 mg/day improves estimated FTP and maximum power performance in amateur cyclists.

## 1. Introduction

Hesperidin is a flavonoid found mainly in citrus fruits [1], reaching high concentration in sweet orange (*Citrus sinensis*) [2]. Due to its chemical structure, including a chiral carbon (C-2), hesperidin can be present as S or R isomer (Figure 1). 2S-Hesperidin is the predominant natural form in citrus fruits [3], but industrial processing leads to the transformation of the natural S isomer into the R isomer (Figure 1) [4]. The bioavailability of the two isomers is different, for instance a 5.2-fold higher efficiency in the glucuronidation has been observed for S-hesperetin compared to R-hesperetin in vitro, without any significant change in the sulfonation kinetics [5]. Clinical trials have demonstrated the therapeutic effects of hesperidin and its metabolites in various diseases (e.g., neurological and psychiatric disorders, cardiovascular diseases, etc.) due to its anti-inflammatory properties, antioxidants, inhibition of fat accumulation, improvement in glucose homeostasis and insulin sensitivity [6,7,8,9]. In view of its effects, the pharmaceutical and nutritional industries have extensively marketed hesperidin. However, little attention has been paid to the effects of hesperidin on exercise performance.

Regarding performance, only one acute effects study in humans has investigated 2S-hesperidin [11]. This study showed that after ingesting one single 500 mg dose of either 2S-hesperidin 5 h before the test, trained cyclists significantly improved average power (2.3%), maximum speed (3.2%) and total energy (∑ 4 sprint test; total work) (2.6%) with 500 mg hesperidin supplementation in the best sprint out the four repeated sprint test (4 × 30 s all-out sprints with 5 min of rest between sprints). No significant changes were observed in any of these variables with placebo.

In humans, chronic supplementation of hesperidin has also been studied. Pittaluga et al. [12] investigated the effect of 250 mL of red-orange juice, which has a high content of hesperidin, on exercise performance (incremental test) in healthy, trained older women. Following four weeks of consumption of ROJ (3 per day), these older women significantly increased their work capacity by 9.0% compared to placebo (−1.5%). Another chronic study evaluated the effect of a four-week supplementation of 2S-hesperidin (500 mg/day) in trained cyclists and observed significant increases in average power output (14.9 W = 5.0%) in a 10 min time-trial test on a cycle ergometer, whereas those that consumed placebo had a non-significant increase in average power output (3.8 W = 1.3%), moreover, differences were found when comparing the groups [13]. In addition, another performance-enhancing mechanism has been observed in other substances, such as menthol or capsaicin (polyphenols) through taste, but this pathway has not been explored with 2S-hesperidin [14].

The effect of long-term intake of hesperidin has also been investigated in animal studies. Biesemann et al. [15] observed that six weeks of hesperetin supplementation (main metabolite of hesperidin) (50 mg·kg^−1^·d^−1^) improved running performance by 28.8% (exercise time until exhaustion) compared to placebo in aged mice. This study also found an improvement in endogenous antioxidant enzymes, such as reduced glutathione (GSH), oxidized glutathione (GSSG) and GSH:GSSG ratio. De Oliveira et al. [16] found that four weeks of hesperidin consumption (100 mg/kg body mass) enhanced the antioxidant capacity in the continuous swimming group (183%) and decreased the lipid peroxidation (TBARS) in the interval swimming group (−45%) compared to placebo in rats. In the same line, a recent study in trained animals reported that intake of hesperidin for four weeks improved performance and prevented immune alterations induced by exhausting exercise compared to placebo [17]. Recently, one parallel-group study has shown improvements in the time until exhaustion (58%) on maximal exercise test at 3 weeks of a 5-week chronic supplementation of 2S-hesperidin (200 mg/kg), but not in placebo group (with differences between groups) [18]. In the same study, an enhancement of the antioxidant state was observed (superoxide dismutase (SOD), glutathione peroxidase (GPx)) in the lymphoid and hepatic tissue after the test until exhaustion in the rats that consumed 2S-hesperidin compared to placebo.

Hesperidin strongly increases intracellular ATP compared to the AMP-activated protein kinase (AMPK) activator 5-Aminoimidazole-4-carboxamide ribonucleotide (AICAR), even when AICAR concentration has been increased by 10-fold (100 µM) [15]. In addition, hesperetin (10 µM) has been shown to increase intracellular ATP by 33% and mitochondrial spare capacity by 25%, as well as establish an antioxidant state [15]. Based on the understanding behind the mechanism of hesperidin in vitro, as well as the evidence presented above, hesperidin is a good candidate for improving performance.

Currently, some animal trials have shown that chronic hesperidin supplementation can improve performance, but in humans the evidence is weak and more research is needed. We hypothesised that chronic intake of 2S-hesperidin would improve performance at submaximal and maximal exercise intensities. Therefore, the main aims of this study were to examine the chronic effects of 2S-hesperidin (500 mg, Cardiose^®^) supplementation on: (1) power production at FatMax, ventilatory threshold 1 and 2 (VT1 and VT2) and maximum power in an incremental test (high aerobic component), and (2) maximum absolute and relative power during a Wingate test (high anaerobic component). The secondary objective was to evaluate whether hesperidin supplementation modified metabolic (O_2_ and CO_2_) and energy substrate (carbohydrates and fats) markers during a step test that could explain a possible enhancement in performance.

## 2. Methodology

### 2.1. Participants

Forty healthy, male amateur cyclists participated and completed the study. All the participants had to meet the following inclusion criteria: 18–55 years, BMI of 19–25.5 kg·m^−2^, at least 3 years of cycling experience and training for 6–12 h·wk^−1^. Volunteers were excluded if they: (a) were smokers or regular alcohol drinkers, (b) had a metabolic, cardiorespiratory, or digestive pathology or anomaly, (c) had an injury in the prior 6 months, (d) were supplementing or medicating in the prior 2 weeks and/or (e) had non-normal values in the blood analysis parameters. First, participants were informed about the procedures, and a signed informed consent was obtained. The study was conducted according to the guidelines of the Helsinki Declaration for Human Research [19] and was approved by the Ethics Committee of the Catholic University of Murcia (CE091802).

### 2.2. Study Design

A double-blind, parallel, and randomized experimental design was performed. Randomization was performed using computer software (Randomizer) to assign codes to the groups established in this study [20]. Participants were divided into two groups: experimental (2S-hesperidin; *n* = 20) and control (Placebo; *n* = 20). Total distance of usual training was balanced to make it similar between groups (Table 1). Participants consumed two capsules of 250 mg at the same time of either 2S-hesperidin (500 mg) (Cardiose^®^, produced by HTBA (HealthTech BioActives—Murcia, Spain)) or placebo (microcellulose) for 8 weeks. Specifically, Cardiose^®^ is a natural orange extract that, due to its unique manufacturing process, maintains most of the natural hesperidin isomeric form (NLT 85% 2S-hesperidin). The placebo supplements were also in capsulated form and similar in appearance to the 2S-hesperidin capsule. Cyclists were instructed to take the supplement along with breakfast and to continue their usual diet and training schedule. Subjects in both groups were instructed not to consume foods high in citrus flavonoids (grapefruit, lemons, or oranges) for 5 days prior to and during the study. This was verified by diet recalls records.

### 2.3. Procedures

Participants visited the laboratory on seven occasions. Visit 1 consisted of a medical examination, blood extraction to determine health status and a familiarisation session with the Wingate test. When urine samples were collected on visit 2 in the fasted state, both groups consumed the supplements under the supervision of an investigator, which was followed by a standardized breakfast. On visits 2 and 5, a 24-h diet recall and a Wingate test were performed. On visits 3 and 6, another 24-h diet recall was conducted, followed by an incremental test until exhaustion on a cycle ergometer. On visits 4 and 7, the 24-h diet recall was repeated, and participants performed a step test on the cycle ergometer (Figure 2 and Table 2). Prior to each testing session (visits 2, 3, 4, 5, 6, and 7), a standardized breakfast (557.7 kcal) composed of 95.2 g of carbohydrates (68%), 18.9 g of protein (14%) and 11.3 g of lipids (18%) was prescribed by the sport nutritionist.

### 2.4. Testing

#### 2.4.1. Medical Exam

A medical examination, performed by the research centre’s medical doctor and including health history, resting electrocardiogram and examination (auscultation, blood pressure, etc.), was used to confirm that the volunteer was healthy enough to be enrolled in the study.

#### 2.4.2. Incremetal Test

An incremental step with a final ramp test was performed on a cycle ergometer (Cyclus 2, RBM Elektronik-Automation GmbH, Leipzig, Alemania) using a metabolic cart (Metalyzer 3B. Leipzig, Germany) to determine maximal fat oxidation zone (FatMax), VT1 and VT2 and maximal oxygen consumption (VO_2max_). Participants began cycling at 35 W for 2 min, increasing then by 35 W every 2 min upon attainment of RER > 1.05, participants completed a final ramp 35 W/min until volitional exhaustion. To ensure VO_2max_, at least 2 of the following criteria had to be achieved: plateau in the final VO_2_ values (increase ≤2.0 mL·kg^−1^·min^−1^ in the 2 last loads), reaching maximal theoretical HR ((220—age)·0.95), RER ≥ 1.15 and lactate ≥ 8.0 mmol·L^−1^. VT1 was determined using the criteria of an increase in VE·VO_2_^−1^ (VE = pulmonary ventilation) without further increase in VE·VCO_2_^−1^ and departure from the linearity of VE, whereas VT2 corresponded to an increase in both VE·VO_2_^−1^ and VE/VCO_2_^−1^ [21,22]. All VT1 and VT2 assessments were made by visual inspection of graphs in which were time-plotted against each relevant respiratory variable measured during testing. Ventilatory thresholds were obtained using the ventilatory equivalents method described by Wasserman [23]. FTP was defined as the highest average power output (PO) that can be maintained for 1 h [24]. The estimated functional threshold power (FTP) was calculated using the following equation [25]:FTP (W) = Pmax (W) × 0.865 − 56.484

#### 2.4.3. Step Test

Step test was performed on a cycle ergometer (Cyclus 2, RBM Elektronik-Automation GmbH, Leipzig, Alemania) using a metabolic cart (Metalyzer 3B. Leipzig, Germany) (maximal error: 2%; in power values <100 W) and applying the power output values resulting from the incremental test (FatMax, VT1 and VT2). Participants exercised continuously from FatMax (W) to VT1 (W) and to VT2 (W) for 10 min at each step without rest between them. Cardiorespiratory variables (oxygen consumption (VO_2_), relative oxygen consumption to body mass (VO_2_R), carbohydrate oxidation (CHO), fat oxidation (FAT) and cycling efficiency = (work/energy expenditure) × 100) [26] were determined for each metabolic zones.

#### 2.4.4. Wingate Test

In visit 1 a familiarisation session was performed for this test. Prior to the Wingate test (WAnT), participants warmed up on a cycle ergometer for 10 min at 50 W. The WAnT consisted of an all-out, 30-s sprint on a cycloergometer (Monark Ergomedic 894E Peak Bike, Vansbro, Sweden). Breaking resistance was held constant at 7.5% of each individual’s body mass [27]. All participants were verbally encouraged to pedal as fast as possible during the entire sprint. The anaerobic capacity (non-oxidative) was determined by obtaining the absolute and relative (i.e., to body mass) peak power, initial absolute and relative power, power at maximum speed, time at peak power and time at maximum speed. Participants were familiarized with the WAnT on the same day as the medical exam.

#### 2.4.5. Blood Samples

For blood analytics, two samples were taken, namely one in a 3-mL tube with ethylenediaminetetraacetic acid (EDTA) and another in a 3.5-mL tube with polyethene terephthalate (PET). Red blood cell count was carried out in an automated Cell-Dyn 3700 analyser (Abbott Diagnostics, Chicago, IL, USA) using internal (Cell-Dyn 22) and external (Program of Excellence for Medical Laboratories-PEML) controls. Values of erythrocytes, haemoglobin, haematocrit, and haematimetry indexes were determined. These data were used to verify the health status of the subjects and were not included in the study.

#### 2.4.6. Urine Samples

Main hesperidin metabolites were analysed in participants’ urine. Urine samples, corresponding to the collection of urine 24 h before (V2) and after (V7) the supplementation in both groups for each participant, were frozen in liquid nitrogen after collection and thawed for its analysis. For analysis, 50 µL of urine were mixed with 100 µL of water with 1% formic acid containing the internal standard. Then, the mixture was injected into LC-MS/MS (UHPLC 1290 Infinity II Series coupled to a QqQ/MS 6490 Series Agilent Technologies, Sta. Clara, CA, USA). Metabolites were quantified by external standard calibration, using rac-Hesperetin-d3 as the internal standard).

### 2.5. Statistical Analysis

Statistical analysis was carried out using IBM Social Sciences software (SPSS, v.21.0, Chicago, IL, USA). Data are presented as mean ± SD. Levene and Shapiro–Wilk tests were performed in order to check for homogeneity and normality of the data, respectively. Depending on the normality and homogeneity outcomes obtained, paired T-test or Wilcoxon signed-rank test were carried out to examine within-group pre-post differences. Likewise, between-group comparison was calculated using ANCOVA test or Mann–Whitney U test, using pre-test values as covariates (to eliminate any possible bias caused by the initial level of each group in the different dependent variables). Partial eta squared (ηp2) was calculated as effect size for between-group comparisons. Partial eta square thresholds were used as follow: <0.01, irrelevant; ≥0.01, small; ≥0.059, moderate; ≥0.138, large [28]. Furthermore, the step test data analysis was done using repeated measures T-test to obtain within-group differences when comparing the different time points. Relationships between levels of excreted hesperidin metabolites in urine and other evaluated parameters were analysed using Pearson correlation analysis (r). Significance level was set at *p* ≤ 0.05. Cohen’s d effect sizes (ES) (95% confidence interval) were calculated for comparisons between groups. Threshold values for ES statistics were as follows: >0.2 small, >0.5 moderate, >0.8 large [28].

## 3. Results

### 3.1. Hesperidin Metabolites Urine

Different hesperidin metabolites, mainly hesperetin glucuronides and sulfates, were analyzed in the urine of the participants after 2S-hesperidin intake. The main metabolite detected was hesperetin-3-glucuronide, representing 78.9 ± 5.0% (n = 20) of the total, while hesperetin-7-glucuronide and hesperetin-7-sulfate made up 6.9 ± 2.9% (n = 20) and 14.7 ± 4.1% (n = 20) of the excreted metabolites. Despite the similarities in the excreted metabolites profile, a large interindividual variability was observed in the excreted amount, with hesperidin metabolites ranging from 2.3 to 37.5 μmol. These differences between subjects indicate differences in the absorption and excretion of hesperidin, which have been previously reported [29].

### 3.2. Incremental Test

Figure 3 shows the pre- and post-intervention values and changes in VT1 and VT2 power, estimated FTP and maximum power achieved during the incremental test.

At VT1, there was no differences in pre-post power neither in 2S-hesperidin group (−3.7% = −6.0 W; *p* = 0.437) nor in Placebo group (3.4% = 5.3 W; *p* = 0.453), without differences in VT1 power changes between groups (*p* = 0.423; ηp^2^ = 0.017; ES = 0.35). At VT2, there was no differences pre-post in power output in Placebo (−3.1% = −8.9 W; *p* = 0.264), and no changes were observed in 2S-hesperidin group (1.0% = 2.9 W; *p* = 0.642). Comparison between groups showed no effect (*p* = 0.299; ηp^2^ = 0.029; ES = 0.38).

Interestingly, there were pre-post increases in maximum power (1.9% = 7.4 W; *p* = 0.049) and estimated FTP (2.3% = 6.4 W; *p* = 0.049) in 2S-hesperidin group. However, there were no changes in estimated FTP (−0.9 % = −2.51 W; *p* = 0.387) and maximum power (−0.8% = −2.9 W; *p* = 0.388) after the intervention in the placebo group. Between-group comparisons revealed an effect with the increase in estimated FTP (3.2% = 8.9 W; *p* = 0.042; ηp^2^ = 0.107; ES = 0.68) and maximum power (2.7% = 10.3 W; *p* = 0.042; ηp^2^ = 0.107; ES = 0.68) in 2S-hesperidin group versus placebo.

Additionally, there was a positive correlation between the levels of excreted hesperidin metabolites in urine and the difference in maximum power (r = 0.701; *p* < 0.001) and estimated FTP (r = 0.725; *p* < 0.001) in the supplemented group.

### 3.3. Step Test

At FatMax, there was a pre-post decrease in fat oxidation (FAT) (*p* = 0.007) and efficiency (*p* = 0.010) in the Placebo group, whereas the 2S-hesperidin supplemented group showed no changes in these parameters (Table 3). No differences were found for between-group comparisons in FAT (*p* = 0.125; ηp^2^ = 0.084; ES = 0.59).

At VT1, there was a pre-post increase in carbohydrate oxidation (CHO) (*p* = 0.020) and a decrease pre-post in fat oxidation (*p* = 0.003) in Placebo group, but no changes were observed in 2S-hesperidin (Table 3). No changes were found between groups in CHO (*p* = 0.314; ηp^2^ = 0.028; ES = 0.57) and FAT (*p* = 0.205; ηp^2^ = 0.044; ES = 0.53).

After the supplementation period, there was a decrease in VO_2_ (L/min) (−8.3%; *p* = 0.002) and VO_2_R (ml/kg/min) (−8.9%; *p* = 0.002) at VT2 in Placebo group, in contrast to 2S-hesperidin, which showed no changes (Table 3). Between-group comparison showed a trend towards a decrease (*p* = 0.074; ηp^2^ = 0.084; ES = 0.67) in VO_2_R (mL/kg/min) for placebo versus 2S-hesperidin group.

### 3.4. Wingate Test

Table 4 shows the results of the parameters evaluated during the Wingate test prior and after supplementation, which are also summarized in Figure 4.

In the 2S-hesperidin group, there were increases in absolute (4.9% = 35.5 W; *p* = 0.001) and relative (4.3% = 0.44 W·kg^−1^; *p* = 0.004) initial power (first five seconds of the test), but no differences between groups. In the experimental group, there was an increase in absolute (6.1% = 49.8 W; *p* < 0.001) and relative (5.6% = 0.64 W·kg^−1^; *p* = 0.001) peak power. Also, there was a trend towards an increase in power at maximum speed (4.4% = 34.0 W; *p* = 0.051) and a descending trend in time at peak power (−18.1% = −641.2 ms; *p* = 0.052) after the supplementation with 2S-hesperidin. No changes were observed in time at maximum speed.

Placebo group showed an increase in absolute (6.1% = 47.2 W; *p* = 0.016) and relative peak power (5.6% = 0.64 W·kg^−1^; *p* = 0.014), and a decrease in time at maximum speed (−13.2% = −929.2 ms; *p* = 0.001). No changes were observed in absolute and relative initial power, power at maximum speed and time at peak power for placebo.

Between-group comparison only reported a trend to decrease in time at maximum speed (−12.5% = −878.4 ms; *p* = 0.059) in Placebo compared with 2S-hesperidin.

## 4. Discussion

The main objective of this study was to evaluate the effects of chronic intake of 2S-hesperidin on non-oxidative/glycolytic and oxidative metabolism and performance markers in amateur cyclists. For this purpose, participants were supplemented for eight weeks with 500 mg 2S-hesperidin, a natural extract of sweet orange (*Citrus sinensis*) which contains hesperidin in its natural 2S form (NLT 85% 2S-hesperidin). Following the eight-week intervention, 2S-hesperidin supplementation led to significant improvements in submaximal and maximal intensity exercise performance in the incremental tests versus placebo. There was a significant decrease in VO_2_R (mL/kg/min) at VT2 in placebo compared with 2S-hesperidin, in the step test.

The bioavailability of hesperidin is a factor that must be taken into account when examining its effectiveness, since the average maximum peak blood plasma concentration occurs after 5–7 h of its ingestion and is almost eliminated post-24 h [30]. However, the excreted metabolites in urine has been shown to reach at maximum levels at post-24 h with continued remnants after 48 h [30]. It is interesting to mention that the area under the curve was more than doubled (0.5 L orange juice; 4.19 µmol h/L vs 1l orange juice; 9.28 µmol h/L) at 24 h when high doses of hesperidin were consumed (1L orange juice = 444 mg hesperidin) [30]. This indicates that high doses increase exposure to the body of 2S-hesperidin metabolites than low doses (222 mg/L). The dose that the cyclists in our study consumed was equivalent to more than one liter of orange juice, with the high carbohydrate load that it entails. The metabolites of hesperidin that appear mainly in the blood are glucuronides (87%) and sulfoglucuronides (13%) [30]. These results are very similar to those found in this study.

Another key factor in the metabolism and absorption of 2S-hesperidin is the intestinal microbiota. In particular, Amaretti et al. [31] established that the species *Bifidobacterium catenulatum* and *Bifidobacterium pseudocatenultum* had the ability to hydrolyze hesperidin, because in their genome they have the gene encoding for the enzyme α-L-rhamnose (limiting enzyme), which contributes to the release of aglycone from certain routine-conjugated polyphenols, such as hesperidin. A recent study suggests that the contradictory finding regarding the intake of hesperidin in humans may be due, in part, to the interindividual variability in its bioavailability, which highly depends on the α-rhamnosidase activity and the composition of the gut microbiota [32]. On the other hand, hesperidin has shown to have a probiotic effect by promoting the growth of some beneficial bacterial species in the colon, the key role being the production of short-chain fatty acids (SCFA) (*Bifidobacterium spp*., *Lactobacillus spp*., or *Akkermansia muciniphila*) [33].

### 4.1. Incremental Test

The results of this study showed an improved performance in eFTP and maximum power (↑ generated power) after chronic intake of 2S-hesperidin compared to placebo in incremental test. These changes are supported by a significant positive correlation between excretion of urinary 2S-hesperidin metabolites and maximum power (r = 0.701) and estimated FTP (r = 0.725). Regarding flavonoid supplementation, a previous study reported a 5% increase in absolute power output in a 10-min time trial (TT) after four weeks of 2S-hesperidin intake (500 mg) in cyclists [13]. Other authors have also reported performance improvements (time until exhaustion ~58%) in animals [17,18]. Currently there are no other studies that analyzing the effects of chronic hesperidin intake on performance. Several authors have reported that hesperidin exerts an antioxidative effect and promotes nitric oxide synthesis in different pathological study models [18,34,35,36,37,38,39]. In a rat model with pleurisy, the antioxidant activity of hesperidin reduced the production of ROS in the liver and increased the liver activities of CAT and SOD [35]. Estruel-Amades et al. [18] observed that five weeks of supplementation with 2S-hesperidin (200 mg/kg three days per week) prevented an increase in ROS and decline in SOD and CAT activity after a test until exhaustion in the thymus and spleen of mice with an intensive training plan. This scavenging activity hesperidin neutralizes reactive oxygen species, such as superoxide anion, generated during conditions of oxidative stress, like intense physical exercise [36]. In particular, citrus flavanones (such as hesperidin and hesperetin) have the ability to modulate cellular antioxidant defenses through the Nrf2-ARE pathway, which regulates gene expression of antioxidant enzymes, such as SOD, CAT, HO-1 and GPx, decreasing intracellular pro-oxidants [40]. In addition, several authors have described a stimulating effect of nitric oxide production after hesperidin supplementation [34,37,38,39], by an increase in endothelial activity NO synthase and gene expression of endothelium nitric oxide synthase. NO can relax human vascular cells (vasodilatation), which leads to improved blood flow during rest and exercise. Vasodilation is a physiological mechanism used not only for the supply of oxygenated blood, but also for the delivery of glucose, lipids, and other nutrients to a variety of tissues [41]. Theoretically, increased blood flow would increase the delivery of O_2_ and nutrients (e.g., amino acids and glucose) to exercising skeletal muscle, thus aiding exercise performance during high intensity (conditions of hypoxia) [42]. These mechanisms may be responsible for performance improvement in eFTP and maximum power in the incremental test in the group that consumed 2S-hesperidin.

Other flavonoids such as quercetin, has also demonstrated to improve the 5 km running performance time (−11.3% quercetin group; −3.9% control group) after its 14 day supplementation (250 mg/d) by trained triathletes [43]. However, a systematic review that included 13 randomized controlled trials found that cocoa-derived flavonoid (epicatechin and catechin, and oligomeric procyanidin) supplementation did not affect performance [44]. Thus, there may be some specificity regarding the type of flavonoid that affects physical performance.

It should be noted that this study was carried out during a period when cyclists are reducing their training and competitions (late September–mid December) which involves training misadaptations (physiological and metabolic changes) [45,46]. These changes may justify the drop in the performance at sub-maximal and maximum intensities for placebo in our study. However, the intake of 2S-hesperidin was not able to prevent the loss of performance at VT1, although it was not significant, but it did maintain performance at VT2 and improve it at eFTP and maximum power. This supports our hypothesis, that the chronic intake of 2S-hesperidin could help generate or maintain adaptations at the mitochondrial level and of the endogenous antioxidant system in a period where the volume and intensity of training is decreasing, as in the conducted study (late September–mid December), maintaining performance levels in high-intensity exercise in amateur cyclists. The fact that 2S-hesperidin has an effect on different physiological mechanisms [6,7,8,9] may be the reason why it cannot maintain performance at low but in high exercise intensities. In line with our hypothesis, the intake of hesperitin in elderly rats (hesperidin metabolite) has been shown to prevent loss of performance by improving mitochondrial and endogenous antioxidant status [15]. The improvements in training adaptations of cyclists who ingested 2S-hesperidin may be due to the ability of this molecule to increase gene expression of the peroxisome proliferator-activated receptor-gamma coactivator 1- α (PGC-1α) and nuclear factor respiratory 2 (NRF2), also, it increased the level of proteins of PGC-1α and of complexes I, III, and IV of the electron transport chain in the mitochondria, in muscle cells (in vitro) [15]. In addition, hesperetin has shown increased activation of AMPK in liver cells [47] and fibroblasts [48]. AMPK is a sensor of cellular energy status that plays a central role in skeletal muscle metabolism, regulating muscle exercise capacity, mitochondrial function and contraction-stimulated glucose uptake [49]. PGC-1α and AMPK are an important transcriptional masters regulators of mitochondrial biogenesis (↑ biogenesis mitochondrial and oxidative capacity) [49,50] and NRF2 which is an essential regulator in the control of cellular redox homeostasis and controls glutathione synthesis (reactive oxygen species (ROS) scavenging) [51]. Modifications in these transcription factors have shown performance improvements in endurance athletes [52]. Therefore, 2S-hesperidin has the ability to promote muscle-level adaptations of endurance athletes, which could improve their performance in competitions.

It has been hypothesized that some molecules with anti-inflammatory and antioxidant activity may interfere with exercise-generated adaptations causing a decline in performance when ingested chronically [53]. Although, there is controversy on this issue, since supplementation of polyphenols, such as quercetin, has been shown to improve performance [54]. With the results obtained in the incremental test, we can say that the chronic intake of 2S-heperidin improves the power generated in eFTP and maximum power that would enhance the performance of endurance athletes for competition, avoiding the loss of performance (eFTP and maximum power) observed in the placebo group due to the loss of adaptations achieved during the cycling post-season. In addition, our results were strengthened by the positive correlations found between performance improvements at eFTP and maximum power with the excretion of metabolites in urine after 2S-hesperidin intake. Therefore, an increase in power production at high intensity is a key factor in cycling performance, which can increase your success in endurance competitions. However, at low intensity exercise levels there were no differences between groups. This could be because, at high intensities, the antioxidant action of 2S-hesperidin could improve performance [16,18], but this capacity does not influence exercises at low intensities where oxidative stress is lower.

### 4.2. Step Test

In the step test, the differences found between 2S-hesperidin and placebo indicate mismatches mediated by the reduction in training volume and intensity over the period of the study [45,46], identified as a decrease in FAT (FatMax and VT1) and a decrease in VO_2_R (mL/kg/min) (VT2) in placebo. These findings were in line with those found in the incremental test, where mismatches to training (↓ generated power at eFTP and maximum power) were also found. In endurance athletes, a 7% (*p* < 0.05) and 16% (*p* < 0.05) decrease in VO_2MAX_ after 21 and 56 days of inactivity respectively has already been described in scientific literature, related to a decrease in systolic volume and decrease in citrate synthase and succinate dehydrogenase in muscle activities [55]. Moreover, a decrease in oxygen consumption values in the ventilatory thresholds and in maximum exercise has been associated with a decrease in power outputs in professional cyclists after three weeks of cycling competition [56]. In the detraining process could also be involved the loss of oxidative capacity mediated by the reduction of PGC-1α (↓ mitochondrial content) [15]. Therefore, it is normal that after a period of detraining there are changes in different physiological-biochemical markers that lead to a loss of performance in athletes.

However, the 2S-hesperidin group maintained the oxidation of fats at FatMax and VT1, without decreasing the oxygen consumption in VT2. Similarly, a treatment with low doses of (-)-epicatechin (flavonoid) has shown an attenuation of training losses (14 d of detraining) in skeletal muscle capillarity and bioenergetics achieved after five weeks of resistance training [46]. This suggests a similar effect of both molecules in preventing the physiological changes produced by detraining. In addition, hesperidin (0.5 mmol·kg^−1^ of body mass) intake has been shown to be effective in reducing the accumulation of body fat mass, glucose levels and blood lipids in rats fed a high-fat diet [57]. The possible pathways used by chronic intake of 2S-hesperidin to decrease physiological changes derived from detraining would be related to the modulating gene components, such as AMPK and PGC-1α [15,47,48], which control energy production, utilization of metabolic substrates (fats and carbohydrates), mitochondrial biogenesis and oxidative capacity [49,50]. Our results suggest that chronic intake of 2S-hesperidin may prevent the decrease in VO_2_R (mL/kg/min) (VT2) that is associated with a decrease in the ability to produce power in cyclists, and a drop in FAT (FatMax and VT1), increasing carbohydrate utilization at moderately low intensities, which could anticipate fatigue in subsequent high-intensity work, such as in a cycling competition.

### 4.3. Wingate Test

The results obtained in Wingate test (high anaerobic component) after intake 2S-hesperidin showed an improvement in both initial power absolute and relative when compared to placebo. On the other hand, both groups improved both power variables for a 30 s sprint (Wingate test), without differences when comparing the groups. Currently, there are no other studies that have evaluated the chronic intake of 2S-hesperidin using a Wingate test. Martínez et al. [11] observed improvements in average power (2.3%) and maximum speed (3.2%) during a repeated 30-s sprint test in amateur cyclists following an acute intake of 2S-hesperidin. However, there are no previous studies that have evaluated the effect of chronic hesperidin intake on maximum anaerobic capacity (non-oxidative). In addition, combined intake of mangiferin and luteolin (polyphenols) for 15 days has also displayed improvements in average power (5.0%) during a Wingate [58].

In the short maximum effort tests, some of the changes can be explained by an initial learning effect, followed by a typical variation within the test(s) [59]. Considering that the significant differences between the two experimental conditions have been small in the measurements evaluated in the Wingate test, it should be taken into account that in this type of trial they may be susceptible to the effects of placebo, nocebo or Hawthorne [60,61]. Intra-individual variability and therefore the probability of committing a type one error was further reduced by assessing study subjects at approximately the same time of day, thus avoiding effects of the circadian system about physiological, psychological, and molecular mechanisms in the body, resulting in varying physical performance over the day [62]. We consider that in this type of test (Wingate) familiarisation can have an important effect on the final results, therefore, for future research we will introduce a comparison between the values obtained in familiarisation and the placebo, in order to observe variations that can affect the final result or when comparing experimental groups taking into account the variability of the test [63].

One limitation of our study is the lack of having muscle biopsies to examine the possible mechanisms that could explain these improvements due to financial restrictions. They could have provided valuable.

### 4.4. Practical Applications

The data found in this research shows how chronic intake of 2S-hesperidin enhances performance in FTP and maximum power. Advances in these areas of intensity are crucial for improving results in cycling competitions. Furthermore, as observed in the step test, 2S-hesperidin has the ability to maintain oxygen consumption in VT2 and fatty acid oxidation levels in FatMax and VT1, in periods with a decrease in training exercise volume and intensity (i.e., this study was conducted in the off-season). Given the effects reported by 2S-hesperidin, sports nutritionists would have other ergogenic aids available to improve the performance of their athletes. In this period, cyclists had decreased the volume and intensity of training with respect to other periods of the year. This is an important aspect to consider when comparing our results with other studies, as the outcomes could be different due to the volume and intensity of usual training during the testing time period.

## 5. Conclusions

Supplementation with 2S-hesperidin for eight weeks promotes an improvement in estimated FTP and maximum power in amateur cyclists during an incremental test. Furthermore, the supplementation with 2S-hesperidin can prevent a drop in VO_2_R (VT2) and FAT (FatMax and VT1) in step test on training periods with less volume and load. These findings support the use of 2S-hesperidin as a natural new ergogenic aid, which can help cyclists improve both their aerobic performance.

## Figures and Tables

**Figure 1 nutrients-12-03911-f001:**
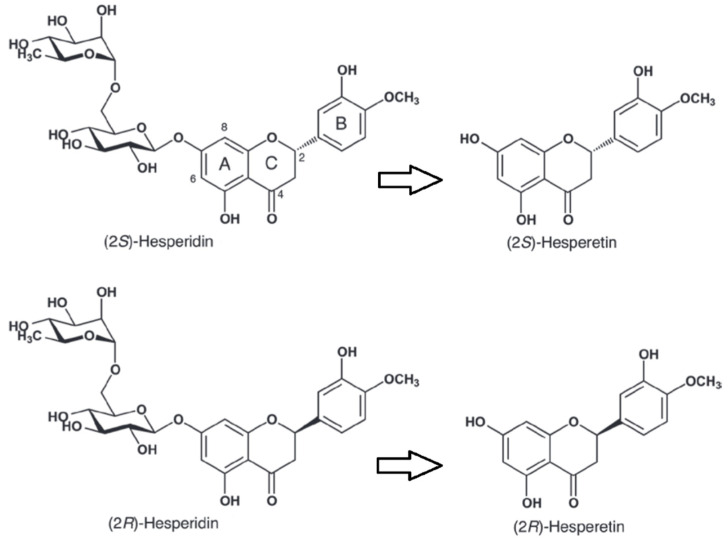
Structure of hesperidin enantiomers S and R and their metabolites hesperetin, produced by the intestinal microbiota. Modified from Li et al. [10].

**Figure 2 nutrients-12-03911-f002:**
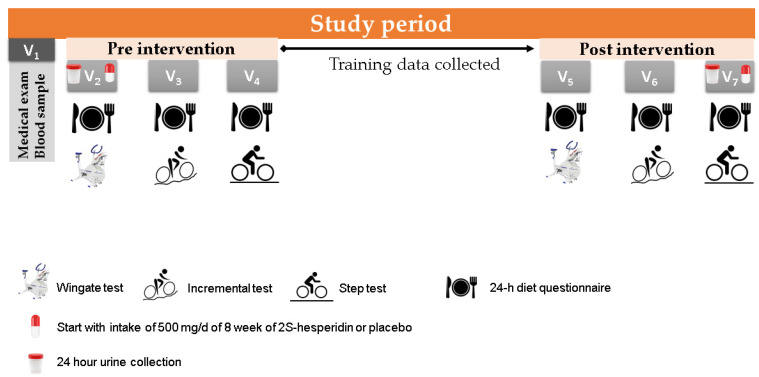
Study planning with explanation of the different visits (V 1–7).

**Figure 3 nutrients-12-03911-f003:**
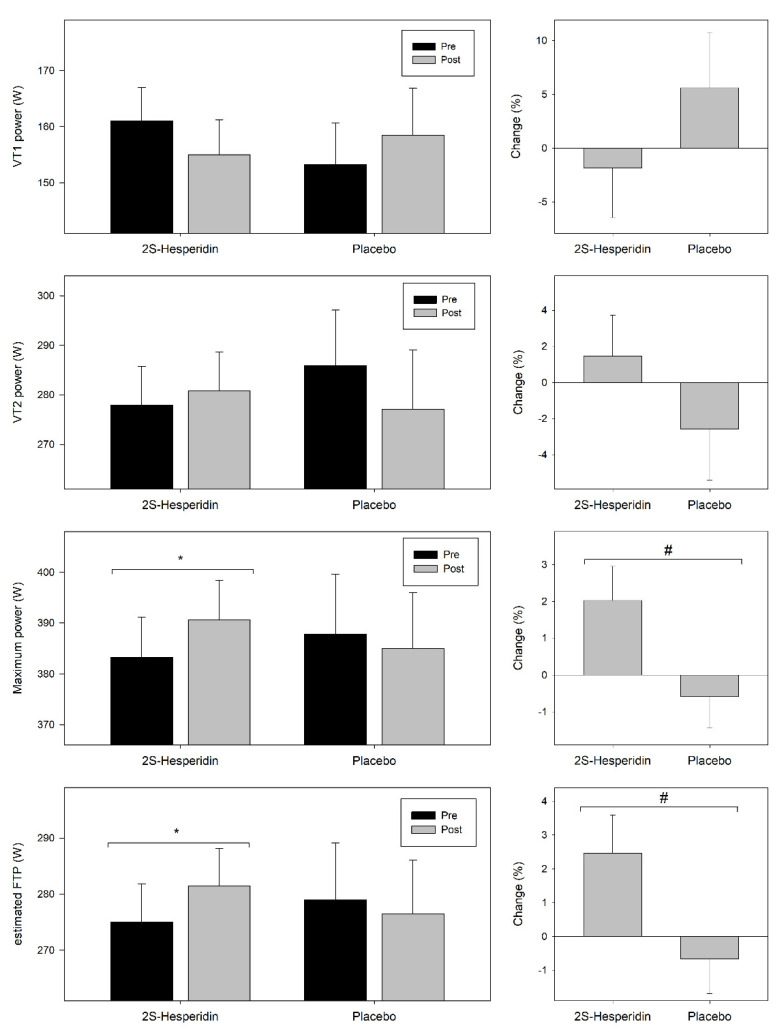
Changes in ventilatory 1 (VT1) power, ventilatory threshold 2 (VT2) power, estimated functional threshold power (FTP) and maximum power during the incremental test. Values are mean ± SE. * Within-group significant changes (*p* ≤ 0.05). # Between group significant changes (*p* ≤ 0.05).

**Figure 4 nutrients-12-03911-f004:**
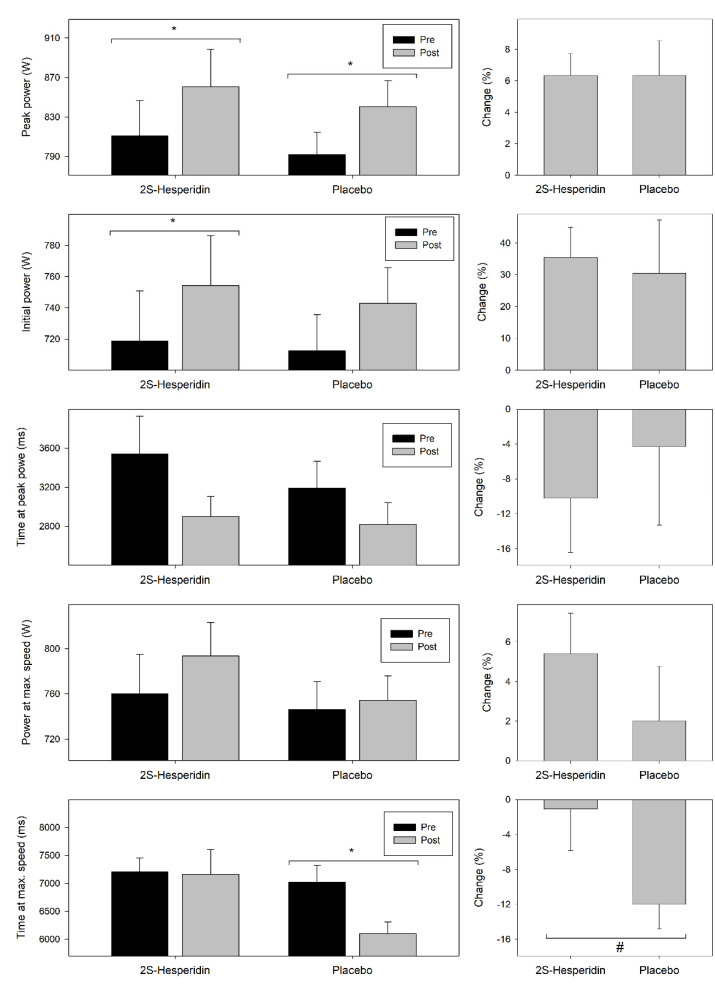
Changes in parameters evaluated during the Wingate test prior and after supplementation. Values are mean ± SE. * Within-group significant changes (*p* ≤ 0.05). # Between group trend t significant changes (*p* = 0.05–0.010).

**Table 1 nutrients-12-03911-t001:** Baseline general characteristics and training variables of participants.

	2S-Hesperidin	Placebo	*p*-Value
Age (years)	35.0 (9.20)	32.6 (8.90)	0.407
Body mass (kg)	71.0 (6.98)	70.4 (6.06)	0.773
Height (cm)	175.3 (6.20)	176.5 (6.10)	0.541
BMI (kg·m^−2^)	23.1 (1.53)	22.6 (1.43)	0.292
BF (%)	8.9 (1.63)	9.0 (1.64)	0.803
VO_2MAX_ (L·min^−1^)	3.99 (0.36)	3.98 (0.63)	0.971
VO_2MAX_ (mL·kg^−1^·min^−1^)	57.5 (6.97)	57.9 (9.53)	0.880
HR_MAX_ (bpm)	184.9 (11.11)	183.2 (8.68)	0.593
VT1 (%)	50.9 (5.63)	50.0 (4.78)	0.610
VT2 (%)	84.9 (5.85)	84.1 (5.70)	0.644
Training variables	2S-Hesperidin	Placebo	*p*-value
Total distance (km)	1121.12 (534.99)	1082.43 (810.46)	0.868
HR_AVG_ (bpm)	144.76 (8.88)	137.48 (13.11)	0.067
W_AVG_ (W)	174.9 (15.79)	163.5 (32.49)	0.435
RPE	6.34 (0.82)	6.33 (1.16)	0.975

Values are expressed as mean (SD). BMI = body mass index; BF = body fat; VO_2máx_ = maximum oxygen volume; VT1 = ventilatory threshold 1 (aerobic); VT2 = ventilatory threshold 2 (anaerobic); Total distance = of all the training sessions carried out during the study period; HRavg = average heart rate of all the training sessions carried out during the study period; Wavg = average power output of all training sessions during the study period.

**Table 2 nutrients-12-03911-t002:** Between-group comparisons in dietary intake of cyclists.

	Pre-Intervention			Post-Intervention		
	2S-Hesperidin	Placebo	*p*-Value	2S-Hesperidin	Placebo	*p*-Value
Kcal	2163.6 (519.02)	2100.2 (515.77)	0.708	1974.1 (377.97)	2133.5 (437.98)	0.237
Kcal/BM	31.1 (9.34)	30.2 (8.71)	0.768	27.9 (6.53)	30.3 (6.46)	0.249
CHO (g)	245.7 (73.46)	222.0 (69.68)	0.312	216.6 (63.47)	248.3 (58.15)	0.117
CHO/BM	3.5 (1.31)	3.2 (1.14)	0.416	3.1 (1.08)	3.5 (0.94)	0.173
PRO (g)	113.5 (25.21)	115.2 (25.37)	0.837	109.0 (23.05)	101.5 (23.67)	0.332
PRO/BM	1.6 (0.41)	1.7 (0.48)	0.778	1.5 (0.35)	1.5 (0.42)	0.596
LP (g)	80.8 (27.24)	83.5 (23.65)	0.739	71.5 (17.61)	71.6 (18.89)	0.985
LP/BM	1.2 (0.45)	1.2 (0.37)	0.758	1.0 (0.27)	1.0 (0.29)	0.823

Values are expressed as mean (SD). Kcal = kilocalories; CHO = carbohydrates; PRO = protein; LP = lipids; BM = body mass. The mean values correspond to the average of all 24-h diet recall data collected at pre-intervention (visits 2, 3 and 4) and post-intervention (visits 5, 6 and 7).

**Table 3 nutrients-12-03911-t003:** Changes in metabolism, energy substrate, energy and energy efficiency in FatMax, ventilatory threshold 1 (VT1) and ventilatory threshold 2 (VT2) during the step test.

	2S-Hesperidin			Placebo				
	Pre-Intervention	Post-Intervention	*p*-Value	Pre-Intervention	Post-Intervention	*p*-Value	ηp^2^	ES
FatMax								
VO_2_ (L·min^−1^)	2.23 (0.50)	2.02 (0.37)	0.063	2.27 (0.48)	2.10 (0.57)	0.151	0.005	0.08
VO_2_R (ml·kg^−1^·min^−1^)	31.45 (6.17)	28.54 (5.43)	0.060	32.40 (6.82)	29.51 (6.99)	0.100	0.003	0.00
CHO (g·min^−1^)	2.20 (0.58)	2.01 (0.37)	0.169	2.20 (0.50)	2.27 (0.56)	0.521	0.090	0.47
FAT (g·min^−1^)	0.29 (0.90)	0.26 (0.14)	0.247	0.32 (0.14)	0.21 (0.14)	0.007	0.064	0.59
Efficiency (%)	26.68 (2.95)	26.05 (3.90)	0.411	26.94 (2.79)	24.62 (2.27)	0.010	0.064	0.49
VT1								
VO_2_ (L·min^−1^)	2.19 (0.39)	2.10 (0.35)	0.396	2.10 (0.41)	2.09 (0.47)	0.961	0.001	0.17
VO_2_R (ml·kg^−1^·min^−1^)	31.05 (5.34)	29.62 (5.20)	0.357	29.96 (5.84)	29.64 (6.37)	0.824	0.001	0.17
CHO (g·min^−1^)	2.08 (0.47)	2.07 (0.30)	0.974	1.86 (0.47)	2.19 (0.49)	0.020	0.028	0.57
FAT (g·min^−1^)	0.31 (0.10)	0.27 (0.15)	0.184	0.35 (0.12)	0.23 (0.14)	0.003	0.044	0.53
Efficiency (%)	26.55 (2.62)	25.25 (5.38)	0.250	27.49 (3.25)	25.86 (5.85)	0.282	<0.001	0.77
VT2								
VO_2_ (L·min^−1^)	3.49 (0.43)	3.36 (0.41)	0.135	3.63 (0.52)	3.33 (0.54)	0.002	0.039	0.49
VO_2_R (ml·kg^−1^·min^−1^)	49.48 (6.83)	48.25 (6.84)	0.211	51.90 (8.17)	47.29 (7.76)	0.002 ^†^	0.084	0.67
CHO (g·min^−1^)	5.11 (1.18)	5.42 (1.37)	0.349	5.53 (1.45)	5.25 (1.13)	0.369	0.022	0.43
FAT (g·min^−1^)	0.04 (0.08)	0.04 (0.09)	1.000	0.02 (0.06)	0.01 (0.03)	0.334	0.048	0.03
Efficiency (%)	20.58 (3.09)	19.65 (3.37)	0.272	20.15 (2.25)	20.20 (4.30)	0.965	0.009	0.24

Values are mean (SE). VO_2_ = volume of oxygen uptake; VO_2_R = body mass oxygen consumption; FatMax = intensity at which maximum fat oxidation is given; VT1 = ventilatory threshold 1 (aerobic); VT2 = ventilatory threshold 2 (anaerobic); CHO = carbohydrate oxidation; FAT = fat oxidation; efficiency = percentage. The *p*-values refer to intra-group comparisons. There were no significant changes when comparing the groups. The trend towards significance between groups is indicated by a ^†^.

**Table 4 nutrients-12-03911-t004:** Changes in performance parameters in the Wingate test.

	2S-Hesperidin			Placebo				
	Pre-Intervention	Post-Intervention	*p*-Value	Pre-Intervention	Post-Intervention	*p*-Value	ηp^2^	ES
Initial power absolute (W)	718.8 (143.05)	754.3 (143.09)	0.001 *	712.5 (103.46)	743.0 (101.78)	0.084	0.003	0.08
Initial power relative (W)	10.2 (1.82)	10.6 (1.78)	0.004 *	10.1 (1.38)	10.6 (1.29)	0.078	<0.001	0.01
Absolute peak power (W)	810.8 (160.26)	860.6 (170.37)	<0.001 *	792.0 (100.96)	840.2 (118.93)	0.016 *	<0.001	0.02
Relative peak power (W)	11.5 (2.04)	12.1 (2.27)	0.001 *	11.3 (1.37)	11.9 (1.49)	0.014 *	<0.001	0.02
Power at maximum speed (W)	760.0 (156.45)	793.5 (132.23)	0.051 ^†^	746.3 (110.30)	754.3 (96.14)	0.709	0.044	0.30
Time at peak power (ms)	3541.4 (1722.52)	2900.2 (923.99)	0.052 ^†^	3193.4 (1218.48)	2816.9 (1013.54)	0.138	0.001	0.82
Time at maximum speed (ms)	7208.7 (1098.24)	7157.9 (2005.11)	0.888	7024.4 (1347.65)	6095.2 (957.33)	0.001 *	0.119	0.73

Values are mean (SE). * Within-group significant changes (*p* ≤ 0.05). ^†^ Within-group trend to significant changes (*p* = 0.05–0.010).

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
