# Peer review of "Effects of 8 Weeks of 2S-Hesperidin Supplementation on Performance in Amateur Cyclists"

_nutrients, 2020, doi:10.3390/nu12123911_

Round 1
Reviewer 1 Report
Thank you for the opportunity to review this resubmitted version of this manuscript. It's pleasing to see that the authors have taken on board comments from previous reviews.
General comments:
Please check the capitalisation of hesperidin throughout, at present this is inconsistent; I'd suggest it's only capitalised at the start of a sentence.
Anaerobic and aerobic may be amended to non-oxidative/glycolytic and oxidative at the authors' discretion. The terms aerobic and anaerobic are becoming a little dated
Is the fact that your athletes are experiencing detraining effects not a major limitation of this study and the nature of the findings? Instead of asking whether or not hesperidin improves performance, are you not in fact asking whether or not hesperidin prevents the anticipated decline in fitness throughout an off-season period? This would be a highly novel research angle, that may be worth considering further.
Line by line:
21 - please add 'the' before other group
23/24 - what are the metabolic aerobic-anaerobic parameters? This is unclear at present. I'd suggest breaking this down into specific tests instead of broad themes. This will support your reported results.
24/29 - what duration are your Wingate tests?
Figure 1 - interesting choice to include the metabolite hesperitin here too, what's the relevance of this to the present study?
53 - by Martinez et al is not required here, the reference will suffice i.e. [11]
54 - please delete 'or placebo (cross over study)' as this currently reads like both hesperidin and placebo improved performance
59/67 - please consider flipping the order of these comments. The first example discusses hesperidin from food sources. Whilst this is pertinent, it does not naturally follow the previous paragraph. You may also want to consider including the following papers which mention the form of ergogenic aids and their efficacy
Best, R., McDonald, K., Hurst, P., & Pickering, C. (2020). Can taste be ergogenic?. European Journal of Nutrition.
Pickering, C. (2019). Are caffeine’s performance-enhancing effects partially driven by its bitter taste?. Medical hypotheses, 131, 109301.
83 - please remove scientific; this sentence may also sit better after the comments regarding ATP
93 - please amend was to were
102 - I think you can remove the reference to Table 1 here
110 - please add university committee details/ approval number
134 - please amend Tables to Table
Figure 2 - just double check this, as you've still got the rectangular test listed as opposed to step test
146-161 - this is really clear, nice job
172 - is the abbreviation for the Wingate test necessary?
202 - please amend possiblility to possibly, or remove
I don't think you need both a Cohens d and partial eta squared. Both are measures of effect, and are typically used for between group comparisons. It may be best in this instance to stick with the partial eta squared, as this aligns with your ANCOVA approach more readily, although Cohen's d would also work for within group comparisons over time
Figure 3 - nice job on this, great to see the change figures included too
Table 3 - I find it slightly odd how after 8 weeks of training we're seeing a reduction in people's efficiency. I look forward to seeing this explained in the discussion, as this seems highly unusual in a trained group.
Table 4 - please amend your > to < for your absolute power value for Hesperidin
287 - I see you've tried to include all your statistical parameters here, which is great, but it feels a little cluttered. You've seen a moderate difference in CHO oxidation, it may be best to simply state this and report the p value subsequently, to demonstrate non-significant results
297 - with or without the CHO load?
300-318 - whilst this is very interesting, I question the relevance of this section to the present work. Consider including this as part of directions for future research, but if it is to be included there, it needs to be condensed and tied to the present work
344-346 - I like your reasoning here, but isn't NO more likely to be of use to VT1 than VT2 and FTP, especially given that these results (VO2, VO2R etc.) approached significance at VT1 but were not significant at higher intensities?
356 - you still had negative outcomes in the treatment group, as well as the placebo group. Please discuss/ state this too
367 - please remove 'As the scientific evidence shows'
384 - I find it interesting that you've not made more of these correlations, these are really nice results that need to be highlighted more in both the results section and discussion
Rest of discussion not reviewed at present due to extensive nature of changes recommended above
Author Response
Reviewer 1
Thank you for the opportunity to review this resubmitted version of this manuscript. It's pleasing to see that the authors have taken on board comments from previous reviews.
Response: We thank the reviewer for their constructive and helpful feedback on our manuscript. We have replied to each specific comment in the section below and have introduced the corresponding edits into the manuscript using Word track changes.
Please check the capitalisation of hesperidin throughout, at present this is inconsistent; I'd suggest it's only capitalised at the start of a sentence.
Response: Following your suggestion, we have edited this throughout the text.
Anaerobic and aerobic may be amended to non-oxidative/glycolytic and oxidative at the authors' discretion. The terms aerobic and anaerobic are becoming a little dated.
Response: Following your suggestion, we have made the appropriate modifications.
Is the fact that your athletes are experiencing detraining effects not a major limitation of this study and the nature of the findings?
Response: We not believe that this is a major limitation because just as the training can affect athletes differently, so can detraining. Thanks for your comment.
Instead of asking whether or not hesperidin improves performance, are you not in fact asking whether or not hesperidin prevents the anticipated decline in fitness throughout an off-season period? This would be a highly novel research angle, that may be worth considering further.
Response: As you rightly comment, we focus the manuscript’s discussion on how this supplement prevents detraining effects during the off-season period. However, we observed an increase in eFTP performance and maximum power in the 2S-hesperidin group, thus implying performance improvements. Thanks for your comment.
Line by line:
21 - please add 'the' before other group
Response: Amended. Line 22.
23/24 - what are the metabolic aerobic-anaerobic parameters? This is unclear at present. I'd suggest breaking this down into specific tests instead of broad themes. This will support your reported results.
Response: Following your suggestion, we have modified this sentence. Lines 23-25.
24/29 - what duration are your Wingate tests?
Response: The duration of the Wingate test was 30 seconds. We have included the duration in the manuscript. Line 27.
Figure 1 - interesting choice to include the metabolite hesperitin here too, what's the relevance of this to the present study?
Response: When hesperidin is ingested as food, the bacterial flora metabolises it to produce hesperetin, which is the main active metabolite that appears in the blood and urine. This process occurs in the vast majority of polyphenols, where the intestinal flora is a key step in their biological activity. Thus, we believe it is essential that the readers understand that they are different molecules. Thanks for your comment.
53 - by Martinez et al is not required here, the reference will suffice i.e. [11]
Response: As suggested, we have removed this reference. Line 56.
54 - please delete 'or placebo (cross over study)' as this currently reads like both hesperidin and placebo improved performance
Response: As suggested, we have removed these words. Line 57.
59/67 - please consider flipping the order of these comments. The first example discusses hesperidin from food sources. Whilst this is pertinent, it does not naturally follow the previous paragraph. You may also want to consider including the following papers which mention the form of ergogenic aids and their efficacy
Best, R., McDonald, K., Hurst, P., & Pickering, C. (2020). Can taste be ergogenic?. European Journal of Nutrition.
Pickering, C. (2019). Are caffeine’s performance-enhancing effects partially driven by its bitter taste?. Medical hypotheses, 131, 109301.
Response: We believe it is more correct to make the first paragraph about the acute effects of hesperidin intake, and then to comment on the chronic effects on humans in the second paragraph. We believe that a good distinction should be made between acute and chronic intake in order not to confuse the reader. In the paragraphs that follow, we discuss the effects of hesperidin in animals. In addition, we believe that there can be no ergogenic effect through the taste of hesperidin, since, the intake was made by capsule and that there was no intervention regarding taste. As to your suggestion, we have included one of the proposed references. Lines 69-71. Thanks for your comment.
83 - please remove scientific; this sentence may also sit better after the comments regarding ATP
Response: As suggested, we have made appropriate modifications. Lines 96-98.
93 - please amend was to were
Response: Amended. Line 102.
102 - I think you can remove the reference to Table 1 here
Response: Amended. Line 111.
110 - please add university committee details/ approval number
Response: Amended. Lines 119-120.
134 - please amend Tables to Table
Response: Amended. Line 144.
Figure 2 - just double check this, as you've still got the rectangular test listed as opposed to step test
Response: As you have suggested, we have modified Figure 2.
146-161 - this is really clear, nice job
Response: Thank you very much for your comment.
172 - is the abbreviation for the Wingate test necessary?
Response: We have not included any abbreviation about the Wingate test, as it may confuse the reader with another abbreviation such as 'W' (watts).
202 - please amend possiblility to possibly, or remove
Response: Removed. Line 213.
I don't think you need both a Cohens d and partial eta squared. Both are measures of effect, and are typically used for between group comparisons. It may be best in this instance to stick with the partial eta squared, as this aligns with your ANCOVA approach more readily, although Cohen's d would also work for within group comparisons over time
Response: Partial eta squared was introduced because of a previous reviewer’s suggestion. But we decided to also include Cohens d, since some reviewers believe that this statistical method is more appropriate to compare the effect between groups. We would like to leave both Cohens d and partial eta squared. However, if you think it is unnecessary, can eliminate one of them. Thanks for your comment.
Figure 3 - nice job on this, great to see the change figures included too
Response: Thank you very much for your comment.
Table 3 - I find it slightly odd how after 8 weeks of training we're seeing a reduction in people's efficiency. I look forward to seeing this explained in the discussion, as this seems highly unusual in a trained group.
Response: We have discussed this later in the manuscript, contextualizing the moment in which we carried out the study (off-season), since cyclists reduce the volume and intensity of their training, consequently having a decrease in their performance and changes at a metabolic level due to detraining.
Table 4 - please amend your > to < for your absolute power value for Hesperidin
Response: Amended.
287 - I see you've tried to include all your statistical parameters here, which is great, but it feels a little cluttered. You've seen a moderate difference in CHO oxidation, it may be best to simply state this and report the p value subsequently, to demonstrate non-significant results
Response: This paragraph has been removed following the suggestion by other reviewers.
297 - with or without the CHO load?
Response: As we commented in the methodology section, participants ate a standardized breakfast before the tests.
300-318 - whilst this is very interesting, I question the relevance of this section to the present work. Consider including this as part of directions for future research, but if it is to be included there, it needs to be condensed and tied to the present work
Response: As suggested, we have condensed this paragraph, leaving the concepts that are necessary to understand how the intestinal flora can affect the bioavailability and efficacy of 2S-hesperidin. Thanks for your comment.
344-346 - I like your reasoning here, but isn't NO more likely to be of use to VT1 than VT2 and FTP, especially given that these results (VO2, VO2R etc.) approached significance at VT1 but were not significant at higher intensities?
Response: We believe that the delivery of oxygen and nutrients to the muscle may be more compromised at high intensities, therefore, a greater delivery of these would facilitate improved performance. It is not clear to us that we have answered your question. If not, please let us know. Thanks for your comment.
356 - you still had negative outcomes in the treatment group, as well as the placebo group. Please discuss/ state this too
Response: Following your suggestion, we have introduced all results, including the negative findings in the hesperidin group. Lines 375-377 and 381-384. Thanks for your comment.
367 - please remove 'As the scientific evidence shows'
Response: Amended. Line 391.
384 - I find it interesting that you've not made more of these correlations, these are really nice results that need to be highlighted more in both the results section and discussion
Response: These correlations are mentioned in lines 242-244 of the results section, but as you have suggested, we have highlighted them more in the discussion to give strength to our results. Lines 339-340. Thanks for your comment.
Author comment: We appreciate all the comments made on our manuscript, which helped improve it’s quality.

Reviewer 2 Report
Thank you for your responses to my concerns from earlier. I appreciate that you have made some revisions. Below are concerns I still have:
- In the introduction, in some places you clarify whether supplementation with hesperidin changed significantly different compared to supplementation with placebo (when referring to other published studies), but in other places you only point out the change for the supplementation group. It is not clear, therefore, whether this change is due to chance or training or to the supplement.
- In the familiarization session (visit 1), did participants actually complete a Wingate test?
- Why did all participants have the identical breakfast (absolute grams of nutrient) instead of g/kg? Furthermore, dietary intake from the food recalls should be presented both in absolute (g) and relative (g/kg) terms in Table 2.
- I suggest including the units of measure for VO2 (L/min) and VO2R (ml/kg/min, presumably) since I've never seen the abbreviation VO2R.
- Please refer to the supplement throughout your paper as hesperidin and not the brand name to avoid semblance of advertising (ex: line 213, Figure 3, line 281, 461, etc).
- The figures are still extremely small, even though there is room on the page for larger images. Why are they still so small?
- Table 3 is difficult to interpret. It seems that the p values presented are just within-group changes, but the more compelling results would be if there is a significant time by group difference. Regardless, it should be clarified what this is referring to.
- I'm concerned your statistics are inappropriate. You should have just done at ANCOVA to look for time by group differences first and then explored the main effect of group only if warranted. Doing ANCOVAs in addition to t-tests seems to be inappropriate.
- Lines 259-272 ought to be redesigned first to show if there was a significant between-group difference. The way that the increase in power among the supplement group is presented first implies that it is more than the change with the placebo group, but it does not appear that this is actually the case.
- The discussion addresses a number of non-significant changes for no apparent reason. Such changes must have a considerable effect size and be close to reaching statistical significance to merit such mention.
Author Response
Reviewer 2
Thank you for your responses to my concerns from earlier. I appreciate that you have made some revisions. Below are concerns I still have:
Response: We thank the reviewer for their constructive and helpful feedback on our manuscript. We have replied to each specific comment in the section below and have introduced the corresponding edits into the manuscript using Word track changes.
- In the introduction, in some places you clarify whether supplementation with hesperidin changed significantly different compared to supplementation with placebo (when referring to other published studies), but in other places you only point out the change for the supplementation group. It is not clear, therefore, whether this change is due to chance or training or to the supplement.
Response: Amended. Thanks for your comment.
- In the familiarization session (visit 1), did participants actually complete a Wingate test?
Response: Yes, the familiarization with the Wingate test was done and completed in visit 1.
- Why did all participants have the identical breakfast (absolute grams of nutrient) instead of g/kg? Furthermore, dietary intake from the food recalls should be presented both in absolute (g) and relative (g/kg) terms in Table 2.
Response: The breakfast intake was made in absolute grams, covering the requirements of all subjects, and because the indications were easier to follow for the athletes. Considering your suggestion, we have included the food recalls data relative to weight (g/kg). Thanks for your comment.
- I suggest including the units of measure for VO2 (L/min) and VO2R (ml/kg/min, presumably) since I've never seen the abbreviation VO2R.
Response: Amended. We have included the units of measurement in the indicated parameters.
- Please refer to the supplement throughout your paper as hesperidin and not the brand name to avoid semblance of advertising (ex: line 213, Figure 3, line 281, 461, etc).
Response: Amended. Figure 3 was already modified and sent to the editor in the previous review, but we do not know why it was not included in this manuscript. Thanks for your comment.
- The figures are still extremely small, even though there is room on the page for larger images. Why are they still so small?
Response: The figures were modified to improve their quality and sent to the editor in the previous revision, but we do not know why the modified figures did not appear in this manuscript. The figures will be sent back to the editor. Thanks for your comment.
- Table 3 is difficult to interpret. It seems that the p values presented are just within-group changes, but the more compelling results would be if there is a significant time by group difference. Regardless, it should be clarified what this is referring to.
Response: Following your suggestions, we have modified Table 3. Thanks for your comment.
- I'm concerned your statistics are inappropriate. You should have just done at ANCOVA to look for time by group differences first and then explored the main effect of group only if warranted. Doing ANCOVAs in addition to t-tests seems to be inappropriate.
Response: We appreciate your comment. We chose the ANCOVA method based on the existing evidence, which shows that this method has more power than ANOVA for randomised studies although the 2 methods are unbiased doi: 10.1016/j.jclinepi.2006.02.007. In addition, Van Breukelen reported that the required sample size for the ANCOVA is as large as that for ANOVA doi: 10.1152/japplphysiol.00408.2020.
Recently, Salgado et al. found that in general to detect a hypothetical treatment effect of 5%, the ANCOVA approach to analyze differences between treatment groups was more powerful (i.e., more likely to detect a treatment effect) than the ANOVA. However, if the desire was to detect a treatment effect of 20%, the ANOVA and ANCOVA provide similar statistical power doi: 10.1152/japplphysiol.00408.2020. This same author also found that ANCOVA, for pre-post designs offered greater statistical power in endurance performance testing.
On the other hand, the US Food and Drug Administration (2019) has recently issued draft guidance on the topic of baseline covariate adjustment in randomized trials with continuous outcomes. This draft guidance also advocates use of ANCOVA and states that the type I error rate is controlled even when the model mismisspecified. https://www.fda.gov/media/123801/download. Thanks for your comment.
- Lines 259-272 ought to be redesigned first to show if there was a significant between-group difference. The way that the increase in power among the supplement group is presented first implies that it is more than the change with the placebo group, but it does not appear that this is actually the case.
Response: Following your suggestion, we have considered your comments and modified the paragraph. We clarified in line 276-286 that, when comparing the groups, only one trend was reported. Thanks for your comment.
- The discussion addresses a number of non-significant changes for no apparent reason. Such changes must have a considerable effect size and be close to reaching statistical significance to merit such mention.
Response: Following your suggestions, we have eliminated "non-significant changes", thus avoiding confusion for the reader. Thanks for your comment.
Author comment: We appreciate all the comments made on our manuscript, which helped improve it’s quality.

Reviewer 3 Report
General comments
The purpose of this cross-sectional study was to investigate the effects of chronic intake of an orange extract (2S-hesperidin) or placebo on aerobic anaerobic and metabolic performance markers in amateur cyclists. The manuscript is well-written except the results section that should be re-written according to the exact interpretation of the p-value. In general, the p-value is not well interpreted and the results are overinterpreted. Some conclusions do not reflect the data and should be removed from the manuscript. In addition, there is too much reference to the antioxidant properties of hesperidin, which were not measured here. In summary, if the authors interpret their results correctly, the manuscript will be sound and the data could be of interest for sport nutritionists.
Major comments
- The number of decimals reported should be accurately chosen. There is no need to report two decimals for percentages, calories, grams macronutrients or for watts. Please carefully check throughout the manuscript (tables and figures included).
- I suggest to remove the paragraph on animal studies as it is not relevant here. It mainly deals with the antioxidant properties of hesperidin, which was not tested here. In addition, it dilutes the message to be kept from the introduction on human studies.
- Forty subjects were included in the study but it is not mentioned whether a sample size analysis was performed before starting the study. Based on which statistical test was the number of subjects to be included determined? It is important to calculate this before starting the experiment to ascertain that it is not underpowered.
- 2.3. Procedure. The rectangular test is called step test in the detailed description of the test and elsewhere as well. Be consistent in the terms used throughout the manuscript. The same holds true for max test vs incremental test.
- the results section should be thoroughly re-written. If the p-value was <0.05, there is an effect, no need to add that the effect was significant but just mentioning whether there is an increase or a decrease of XX% or by XX-fold. If the p-value was >0.05, there was no effect. It is not correct to state that the effect was not significant. Even worse, it is not allowed to state that the increase or the decrease was not significant as it is, for example, the case in line 226. The whole section needs to be re-written based on this remark.
- the same remark holds true for the abstract that should be re-written accordingly.
- discussion: “In addition, we found a non-significant decrease in FAT at low-moderate intensities (FatMax and VT1), with an increased non-significant with moderate effect in CHO at VT1 in placebo compared to 2S-hesperidin in the step test.” This should be removed from the manuscript as there was no effect. A non-significant decrease means no effect and should be interpreted as such.
- globally, any discussion and conclusion on non-significant changes should be removed from the manuscript. This is a very critical point.
- Lines 323-246 should be removed as they deal whether with animal studies, whether with antioxidant properties of hesperidin, which was not assessed here. In addition, the paragraph starts with performance and antioxidant properties are not directly related to performance, by far.
Minor comments
- 2.2. Study design: “Participants consumed two capsules at the same time of either 2S-hesperidin (500 mg)”. It is not clear whether each capsule weighed 250mg or 500mg.
- line 134: Table 2 and not Tables 2
Author Response
Reviewer 3
General comments
The purpose of this cross-sectional study was to investigate the effects of chronic intake of an orange extract (2S-hesperidin) or placebo on aerobic anaerobic and metabolic performance markers in amateur cyclists. The manuscript is well-written except the results section that should be re-written according to the exact interpretation of the p-value. In general, the p-value is not well interpreted and the results are overinterpreted. Some conclusions do not reflect the data and should be removed from the manuscript. In addition, there is too much reference to the antioxidant properties of hesperidin, which were not measured here. In summary, if the authors interpret their results correctly, the manuscript will be sound and the data could be of interest for sport nutritionists.
Response: We thank the reviewer for their constructive and helpful feedback on our manuscript. We have replied to each specific comment in the section below and have introduced the corresponding edits into the manuscript using Word track changes.
Major comments
- The number of decimals reported should be accurately chosen. There is no need to report two decimals for percentages, calories, grams macronutrients or for watts. Please carefully check throughout the manuscript (tables and figures included).
Response: Amended throughout the text.
- I suggest to remove the paragraph on animal studies as it is not relevant here. It mainly deals with the antioxidant properties of hesperidin, which was not tested here. In addition, it dilutes the message to be kept from the introduction on human studies.
Response: We appreciate your comment. Our intention was never to dilute the message or confuse the reader. In fact, since there are few studies done in humans, we believed that the introduction had to include animals' studies to help the reader understand the possible effects of hesperidin on the various physiological mechanisms on which it acts. We wanted to make clear that not all effects that occur in animals are reproduced in the same way in humans, since certain physiological and metabolic pathways are different. Thanks for your comment.
- Forty subjects were included in the study but it is not mentioned whether a sample size analysis was performed before starting the study. Based on which statistical test was the number of subjects to be included determined? It is important to calculate this before starting the experiment to ascertain that it is not underpowered.
Response: Prior power analysis for the PRE vs. POST difference for a population with a 5% variability in physical parameters, and where a 5% increase is expected (with a power = 0.80), was conducted, and the population needed for a t-test was 32 (16 per group). Considering possible dropouts, we established the sample size to be 20 per group. The study was completed in 40 subjects. Thanks for your comment.
- 2.3. Procedure. The rectangular test is called step test in the detailed description of the test and elsewhere as well. Be consistent in the terms used throughout the manuscript. The same holds true for max test vs incremental test.
Response: Amended. We have changed the term in Figure 2 and in line 211 (rectangular test to step test). Also, max test has been changed to incremental test. Thanks for your comment.
- the results section should be thoroughly re-written. If the p-value was <0.05, there is an effect, no need to add that the effect was significant but just mentioning whether there is an increase or a decrease of XX% or by XX-fold. If the p-value was >0.05, there was no effect. It is not correct to state that the effect was not significant. Even worse, it is not allowed to state that the increase or the decrease was not significant as it is, for example, the case in line 226. The whole section needs to be re-written based on this remark.
Response: Amended. As you have suggested, the results section has been rewritten according to the guidelines indicated. Thanks for your comment.
- the same remark holds true for the abstract that should be re-written accordingly.
Response: Amended. As you have suggested, the abstract has been rewritten according to the guidelines indicated. Thanks for your comment.
- discussion: “In addition, we found a non-significant decrease in FAT at low-moderate intensities (FatMax and VT1), with an increased non-significant with moderate effect in CHO at VT1 in placebo compared to 2S-hesperidin in the step test.” This should be removed from the manuscript as there was no effect. A non-significant decrease means no effect and should be interpreted as such.
Response: Following your suggestion, we have eliminated this phrase.
- globally, any discussion and conclusion on non-significant changes should be removed from the manuscript. This is a very critical point.
Response: As suggested, we have removed the term "non-significant changes" from the discussion and conclusions section. Thanks for your comment.
- Lines 323-246 should be removed as they deal whether with animal studies, whether with antioxidant properties of hesperidin, which was not assessed here. In addition, the paragraph starts with performance and antioxidant properties are not directly related to performance, by far.
Response: We understand your concern, but because there were no human studies that have studied the biochemical-molecular effects, we believed that if we remove all this information from the discussion, we would lose the mechanistic explanation of the action of 2S-hesperidin. And "I say the possible ones" At no time do we intend to extrapolate the findings found in animals to humans. Thanks for your comment.
Minor comments
- 2.2. Study design: “Participants consumed two capsules at the same time of either 2S-hesperidin (500 mg)”. It is not clear whether each capsule weighed 250mg or 500mg.
Response: The subjects consumed 2 capsules of 250 mg hesperidin or placebo every day for 8 weeks. Line 127. Thanks for your comment.
- line 134: Table 2 and not Tables 2
Response: Amended. Line 144.
Author comment: We appreciate all the comments made on our manuscript, which helped improve it’s quality.

Round 2
Reviewer 3 Report
The authors have satisfactorily answered my comments.
This manuscript is a resubmission of an earlier submission. The following is a list of the peer review reports and author responses from that submission.
Round 1
Reviewer 1 Report
Thank you for the opportunity to review this manuscript. Congratulations on completing this study, a very impressive sample size, but I have some concerns about the study, its rationale and how the findings are depicted that must be addressed. I provide a line by line review below.
47 - I'm not sure the terms lipid reducers and insulin sensitisers are appropriate. Is this suggesting an increase in insulin sensitivity following supplementation? Please be more specific here.
57 - please can you clarify whether this was one sprint out of four repeated sprints? Or one trial of four sprints? Also, does total energy refer to total energy expenditure or total work performed? Please also consider wording regarding hesperidin/ Cardiose. At present it is unclear that Cardiose contains hesperidin and in what dose - it may be best to simply leave this as 500mg of hesperidin and not state a brand name
60 - the abbreviation of red orange juice is not needed
69/70 - can you please clarify the size of the improvement in running performance observed?
69-82 - there are a lot of findings presented here, from studies with clearly varying methodologies, doses and outcome measures. I understand your enthusiasm here, as they provide mechanistic support for potential ergogenic effects however, simply listing their significant results in this manner does not provide the reader with sufficient detail. Either make a decision to be more concise, and group studies by dose or outcome measure or take your time, and go into slightly more detail on each study. I would recommend the former, given this is an introduction.
83/88 - this paragraph would be better suited for the discussion, as it contextualises the findings more so than the study's rationale
97/102 - is this paragraph really necessary? I appreciate it somewhat follows on from the above paragraph, but given the above may be better suited to the discussion, it is questionable whether this paragraph adds to the introduction?
109/111 - this wording suggests that you have mentioned extensive amounts of human literature, this would be more appropriate for a review - please reword. Furthermore, you've contradicted yourself, above you discuss maximal efforts following several weeks of supplementation in at least two studies. You also provide in vitro evidence that pertains to maximal performance following longitudinal supplementation. At present, this rationale is flawed. Simplify please.
113/119 - please revisit these sentences and double check wording and word ordering for the purposes of clarity.
118 - I think the term 'rectangular test' may be a mistranslation? I've not come across this term before
Methods
126/127 - were there any supplements in particular that led to exclusion? What blood parameters specifically, were analysed for the purposes of inclusion/exclusion?
Table 1 - its interesting that your athletes HRavg approached being significantly different during exercise despite similarities between all other variables - was this discussed? It is worth noting that these data differ moderately when calculated as an effect size
134/143 - this is really clear, thank you for this. Do you have any other information regarding the contents of the Cardiose? Good to see dose clearly stated.
154 - it is probably worth stating the energy content of the breakfast here, and then providing the macronutrient breakdown. Interesting to note that the nutritionist opted for absolute as opposed to relative intakes of nutrients too.
164/167 - this feels a little rushed. Please double check and add 'a' 'an' etc. accordingly.
167 - please replace the word initialling here. You may want to expand slightly too e.g. upon attainment of RER>1.05, participants completed a final ramp 35W/min until volitional exhaustion.
179/184 - the rectangular test description needs a similar degree of detail to that of the incremental test, please
185/190 - was a warm up completed prior to the Wingate? How was anaerobic capacity calculated?
196 - what other measures were included in haematometric indices? Interesting to see that these values were not included in the study, even at Table 1 given that they may influence cycling performance parameters to an extent
214/215 - by performing multiple t-tests to compare within groups, without a correction you've potentially inflated your risk of type 1 error. In SPSS, at least for your ANOVA, you could perform post-hoc comparisons with an appropriate correction.
Results
I note that you've not included a magnitude of effect for your comparisons, despite this being increasingly recommended within sports science and nutrition, and medical literature. This would be easy to include as a partial eta squared through SPSS
228 - no need to state 'on a cycle ergometer' or to embolden figure name/number in text
237/242 - The reporting of these findings and subsequent interpretation is critical. I am really suspicious of the p values obtained, that are 'just' significant' but are of a smaller percentage magnitude than previously reported non-significant changes. I would also point out to the authors that the magnitude of change is most likely also within the typical error of the testing protocols and or each athlete, and thus should either be interpreted with caution, or considered to be statistically significant but unlikely to be of practical value. On this note, and to facilitate subsequent discussion, it is recommended that the authors include typical error values or similar in the methods section too.
Figure 3/results text (and throughout) - please can you be consistent with your naming of the supplement. You use Cardiose, hesperidin and 2S-hesperidin interchangeably throughout the text (and Figures). Please pick one and use consistently throughout the entire manuscript. My preference would be to opt for either 2S-hesperidin or hesperidin to avoid any conflicts of interest by using a brand name.
Really nice to see change score panels included in Figure 3. On Figure 3, it appears to reinforce my point above regarding meaningful change and error within tests. How can a supplement improve maximal power output at >VO2max, if it has contrasting responses at powers below this threshold? These data either haven't changed 'enough' or the equipment is not sensitive enough to detect the changes brought about by this supplement.
Should there be a discussion of FatMax values and VO2max values obtained from the ramp test, as part of the ramp test results as these inform the rectangular test?
259 - you must introduce this abbreviation (VO2R) earlier in the text, or opt for relative VO2 or VO2REL. You may even wish to simply state VO2 in the table twice and retain units of measurement.
The variables assessed in the Wingate results were not clearly introduced in the methodology. Please amend, as these are interesting.
Given the similarity between responses to placebo and intervention, it wouldn't be unreasonable to suggest that participants have a large learning effect in the Wingate. No familiarisation was mentioned as part of the methods, and we (and others) have previously shown that there can be small to moderate differences in Wingate parameters between familiarisation and intervention trials (Best et al., 2020), suggesting adequate familiarisation is required
Best, R., Temm, D., Hucker, H., & McDonald, K. (2020). Repeated Menthol Mouth Swilling Affects Neither Strength nor Power Performance. Sports, 8(6), 90.
In the interests of time, I feel it perhaps more appropriate to provide comment on the conclusion than the discussion in its entirety. Based upon the data presented, I strongly feel it is unfair to suggest that the supplement in question improves performance to the extent claimed here. The data between placebo and intervention do differ significantly, but often the magnitude of change is likely to fall within the typical error of the test, or intra-individual variability in performance - at least for aerobic measures. Some of these effects may also be explained via learning or Hawthorne effects, especially given placebo consistently performs equally well.
I hope the above has served to strengthen the manuscript in some way.
Author Response
Response to Reviewer 1 Comments
47 - I'm not sure the terms lipid reducers and insulin sensitisers are appropriate. Is this suggesting an increase in insulin sensitivity following supplementation? Please be more specific here.
Response: As suggested, we have modified these terms. Lines 48-49.
57 - please can you clarify whether this was one sprint out of four repeated sprints? Or one trial of four sprints? Also, does total energy refer to total energy expenditure or total work performed? Please also consider wording regarding hesperidin/ Cardiose. At present it is unclear that Cardiose contains hesperidin and in what dose - it may be best to simply leave this as 500mg of hesperidin and not state a brand name
Response: Amended. The sentence refers to the best sprint out of 4 repeated sprints. The total energy refers to the total work performed. Lines 59-60.
60 - the abbreviation of red orange juice is not needed
Response: Amended. Line 63.
69/70 - can you please clarify the size of the improvement in running performance observed?
Response: Amended. Line 73.
69-82 - there are a lot of findings presented here, from studies with clearly varying methodologies, doses and outcome measures. I understand your enthusiasm here, as they provide mechanistic support for potential ergogenic effects however, simply listing their significant results in this manner does not provide the reader with sufficient detail. Either make a decision to be more concise, and group studies by dose or outcome measure or take your time, and go into slightly more detail on each study. I would recommend the former, given this is an introduction.
Response: As you rightly comment, we also think that it would be ideal to group the studies by species, dose and parameter, but this would have resulted in a more expanded introduction. Most of the studies that have used hesperidin were conducted in animals. Very few studies have been performed in humans with this supplement. Therefore, we thought it would be most appropriate to separate the data based on sample species (animal vs. human) so that the results would be interpreted appropriately. We did not wish to mix both sample species (humans and animals) and the corresponding biomarker results to avoid confusion. However, as suggested we have removed the mechanistic support from the introduction to be more concise.
83/88 - this paragraph would be better suited for the discussion, as it contextualises the findings more so than the study's rationale
Response: As suggested, we have moved the paragraph to the discussion. Lines 351-356.
97/102 - is this paragraph really necessary? I appreciate it somewhat follows on from the above paragraph, but given the above may be better suited to the discussion, it is questionable whether this paragraph adds to the introduction?
Response: As suggested, we have moved the paragraph to the discussion. Lines 364-368.
109/111 - this wording suggests that you have mentioned extensive amounts of human literature, this would be more appropriate for a review - please reword. Furthermore, you've contradicted yourself, above you discuss maximal efforts following several weeks of supplementation in at least two studies. You also provide in vitro evidence that pertains to maximal performance following longitudinal supplementation. At present, this rationale is flawed. Simplify please.
Response: As suggested, we have modified the phrase. Lines 115-117.
113/119 - please revisit these sentences and double check wording and word ordering for the purposes of clarity.
Response: As suggested, we have revised sentences to be clearer. Lines 120-123.
118 - I think the term 'rectangular test' may be a mistranslation? I've not come across this term before
Response: As suggested, we have replaced “rectangular test” with “step test” in the whole document.
Methods
126/127 - were there any supplements in particular that led to exclusion? What blood parameters specifically, were analysed for the purposes of inclusion/exclusion?
Response: No type of supplementation could be taken during the study to avoid bias. But in particular those that would have more conflict with our study are the antioxidant supplements. The parameters analyzed in the blood were: haemogram, biochemistry, transaminases, muscle damage and pathologies (Hepatitis and HIV). However, if you consider that this information should be added to the main text, let us know please.
Table 1 - its interesting that your athletes HRavg approached being significantly different during exercise despite similarities between all other variables - was this discussed? It is worth noting that these data differ moderately when calculated as an effect size.
Response: We do not discuss HRavg, since the statistics showed that there were no differences between groups in HR. Also, it should be taken into account that HR can be modified by previous training and other stressors (dehydration, psychological stress, etc.) Ten Haaf T, Foster C, Meeusen R, et al. Submaximal heart rate seems inadequate to prescribe and monitor intensified training. Eur J Sport Sci. 2019;19(8):1082-1091. doi:10.1080/17461391.2019.1571112. Therefore, it would not be the best indicator of internal load. But the average power is, which we have included in table 1, and there were no significant differences between groups.
134/143 - this is really clear, thank you for this. Do you have any other information regarding the contents of the Cardiose? Good to see dose clearly stated.
Response: This is all the information that the company passed on to us.
154 - it is probably worth stating the energy content of the breakfast here, and then providing the macronutrient breakdown. Interesting to note that the nutritionist opted for absolute as opposed to relative intakes of nutrients too.
Response: As you suggested, we have introduced the energy content of breakfast. After breakfast no participant reported any intestinal problems. Line 161.
164/167 - this feels a little rushed. Please double check and add 'a' 'an' etc. accordingly.
Response: As you suggested, we have modified the text. Line 173.
167 - please replace the word initialling here. You may want to expand slightly too e.g. upon attainment of RER>1.05, participants completed a final ramp 35W/min until volitional exhaustion.
Response: As you suggested, we have modified the text. Lines 177-178.
179/184 - the rectangular test description needs a similar degree of detail to that of the incremental test, please
Response: As you suggested, we have included more information in the step test. Lines 191-197.
185/190 - was a warm up completed prior to the Wingate? How was anaerobic capacity calculated?
Response: Before performing the Wingate test the participants warmed up for 10 minutes Lines 200-201. The anaerobic capacity sentence has been modified to avoid confusion. Lines 204-207.
196 - what other measures were included in haematometric indices? Interesting to see that these values were not included in the study, even at Table 1 given that they may influence cycling performance parameters to an extent
Response: The haematometric indices analyzed were erythrocytes, haematocrit and haemoglobin, but as there were no significant differences, we decided not to include them, as the table was already full of data and they were not going to provide anything. As mentioned, the levels of the haemogram indices can influence the performance of cyclists, but we think that what matters is the final expression of performance translated into VO2MAX and W, and in these values, there were no differences between groups at the beginning of the study.
214/215 - by performing multiple t-tests to compare within groups, without a correction you've potentially inflated your risk of type 1 error. In SPSS, at least for your ANOVA, you could perform post-hoc comparisons with an appropriate correction.
Response: Thanks for your comment. Following your suggestions, we have conducted the ANOVA analysis to check possible differences in outcomes. However, the significant and non-significant outcomes do not differ from the previous results reported applying the T-test. Thus, we have not modified the outcomes. However, if you consider that the modification is needed, let us know please.
Results
I note that you've not included a magnitude of effect for your comparisons, despite this being increasingly recommended within sports science and nutrition, and medical literature. This would be easy to include as a partial eta squared through SPSS
Response: Following your suggestion, Eta partial square results has been added. Lines 231.
228 - no need to state 'on a cycle ergometer' or to embolden figure name/number in text
Response: As suggested, we have modified the text.
237/242 - The reporting of these findings and subsequent interpretation is critical. I am really suspicious of the p values obtained, that are 'just' significant' but are of a smaller percentage magnitude than previously reported non-significant changes. I would also point out to the authors that the magnitude of change is most likely also within the typical error of the testing protocols and or each athlete, and thus should either be interpreted with caution, or considered to be statistically significant but unlikely to be of practical value. On this note, and to facilitate subsequent discussion, it is recommended that the authors include typical error values or similar in the methods section too.
Response: Thanks for your comment. According to the cyclo ergometer manual: maximal error: 2% (for power values less than 100 Watt maximally 2 Watt). Cadence error: ± 1 RPM. https://www.cyclus2.com/fileadmin/user_upload/PDFs/cyclus2-brochure.pdf. Line 193.
The metabolic cart (MetaLyzer 3B, Cortex) used in our study has intraclass reliability coefficients of 0.969 (VO2), 0.964 (VCO2), and 0.953 (VE). The Bland-Altman charts revealed slightly lower variability of MetaLyzer 3B measurements compared to other analyser. Therefore, this author concluded that MetaLyzer 3B represents a reliable instrument for exercise testing in sports medicine and research. Meyer T, Georg T, Becker C, Kindermann W. Reliability of gas exchange measurements from two different spiroergometry systems. Int J Sports Med. 2001 Nov;22(8):593-7. doi: 10.1055/s-2001-18523. PMID: 11719895.
Figure 3/results text (and throughout) - please can you be consistent with your naming of the supplement. You use Cardiose, hesperidin and 2S-hesperidin interchangeably throughout the text (and Figures). Please pick one and use consistently throughout the entire manuscript. My preference would be to opt for either 2S-hesperidin or hesperidin to avoid any conflicts of interest by using a brand name.
Response: As suggested, we have removed, "Cardiose" for "2S-hesperidin” in the whole text and figures. We have left the term Cardiose in the methodology section. We would also like to clarify that when we name hesperidin in the text it is because the authors do not specify the formulation, as our hesperidin has a high content of the S-isomer (2S-hesperidin) compared to other hesperidin products, hence the difference in expression.
Really nice to see change score panels included in Figure 3. On Figure 3, it appears to reinforce my point above regarding meaningful change and error within tests. How can a supplement improve maximal power output at >VO2max, if it has contrasting responses at powers below this threshold? These data either haven't changed 'enough' or the equipment is not sensitive enough to detect the changes brought about by this supplement.
Response: We think that because the intake of 2S-hesperidin does not improve performance at intensities below the threshold, we believe this does not condition improvements at high intensities, as there are different metabolic environments (lactate, urea, ammonia, etc.) and there is an increase in the production of free radicals, which can contribute to fatigue, at high intensities compared to low intensities. Therefore, improvements can come from metabolic, biochemical, antioxidant and molecular improvements that have been exposed at the mechanistic level in the discussion. In relation to the equipment used, both the gas analyzer and the ergometer have been used by other authors in their publications. Martinez-Noguera, F.J.; Marin-Pagan, C.; Carlos-Vivas, J.; Rubio-Arias, J.A.; Alcaraz, P.E. Acute Effects of Hesperidin in Oxidant/Antioxidant State Markers and Performance in Amateur Cyclists. Nutrients 2019, 11, doi:10.3390/nu11081898.
Should there be a discussion of FatMax values and VO2max values obtained from the ramp test, as part of the ramp test results as these inform the rectangular test?
Response: We appreciate your question. But we have not discussed FatMax and VO2Max values, as the purpose of each test is different. In the maximal test the objective is to see the power generated in each of the established work zones and in the rectangular the objective is to see the expression of the metabolic values. Therefore, the efforts are not comparable.
259 - you must introduce this abbreviation (VO2R) earlier in the text, or opt for relative VO2 or VO2REL. You may even wish to simply state VO2 in the table twice and retain units of measurement.
Response: As you have suggested, we have introduced in the methodology section the term relative oxygen consumption. Line 195.
The variables assessed in the Wingate results were not clearly introduced in the methodology. Please amend, as these are interesting.
Response: As you have suggested, all the variables measured in the Wingate test have been included in the methodology. Lines 203-205.
Given the similarity between responses to placebo and intervention, it wouldn't be unreasonable to suggest that participants have a large learning effect in the Wingate. No familiarisation was mentioned as part of the methods, and we (and others) have previously shown that there can be small to moderate differences in Wingate parameters between familiarisation and intervention trials (Best et al., 2020), suggesting adequate familiarisation is required
Best, R., Temm, D., Hucker, H., & McDonald, K. (2020). Repeated Menthol Mouth Swilling Affects Neither Strength nor Power Performance. Sports, 8(6), 90.
Response: As you rightly suggest, we also believe that the Wingate test can be influenced by a good learning of the technique. The familiarization with the Wingate test was done on the same day as the medical examination, although it must also be said that most of the subjects who participated in this study were already familiar with this technique, since they had participated in other studies that we carried out previously in our laboratory, since we have a specific database on cyclists.
In the interests of time, I feel it perhaps more appropriate to provide comment on the conclusion than the discussion in its entirety. Based upon the data presented, I strongly feel it is unfair to suggest that the supplement in question improves performance to the extent claimed here. The data between placebo and intervention do differ significantly, but often the magnitude of change is likely to fall within the typical error of the test, or intra-individual variability in performance - at least for aerobic measures. Some of these effects may also be explained via learning or Hawthorne effects, especially given placebo consistently performs equally well.
I hope the above has served to strengthen the manuscript in some way.

Reviewer 2 Report
This is an interesting study about a supplement derived from citrus fruits. It is a long-term (8-week) placebo-controlled study, which strengthens conclusions that can be drawn about the effect of a supplement. Despite this, there are a number of concerns that I have about this study:
- The title does not accurately reflect the study. The title is about a "cyclist" (singular, implying a case study), but the paper is about a group of "cyclists" (plural). Further, both groups (placebo and supplement) improved power in the Wingate test with no significant between group differences for this increase in power. The title is therefore misleading about the efficacy of the supplement vs placebo.
- The figures are very small and the text is hard to read. Can the authors please make these larger?
- Line 50: I suggest rewording this to say "exercise performance".
- Line 54: The studies described in the following paragraph also describe supplementation effects on performance. Therefore, there is NOT only one study that has looked at "physical performance" effects of supplementation.
- Line 87: missing space after "epicatechin"
- Lines 108-109: Are statements presented here also from reference 13?
- Line 112: How do you define "long-term" and "chronic"?
- Line 143: There should be a semi-colon instead of comma.
- Section 2.2: Please expand upon the randomization and placebo. How were participants assigned to their group? Was the placebo matched for visual appearance, taste, and smell?
- Please explain how VO2R (body mass oxygen consumption) differs from VO2 (volume of oxygen uptake).
- Lines 192-193 Missing "a" before describing the tube (ex: a 3-mL tube vs 3 mL tube)
- If the groups were not significantly different from each other in any parameters at baseline, why did you use baseline pre-test variables as covariates?
- Please explain the purpose, validation, and use of the rectangular test in more depth.
- Table 3 defines efficiency as "percentage". What percentage of what?
- Table 4 & Figure 4: I suggest removing the symbol for a trend for a significant change. The reader can see the p value and interpret that for him/herself.
- Lines 293-294: Your conclusion is premature and not based on all of your data. There are plenty of other instances where there were no significant between-group effects (which is really what we're looking at when doing a placebo vs supplement trial). Please discuss the full picture of all of your data, not just significant changes that support your hypothesis.
- Line 347: Should "is" be "in"?
- Line 348: I think to call that an "anti-aging effect" is overstepping the bounds of reasonable interpretation.
- Line 353: "Justified" does not fit the context. Perhaps "demonstrated?
- Lines 363-364: Why the "y"?
- Line 392: Why "Sin embargo, Recently,"?
- Line 472: Missing a space between ROS & One
- The discussion reads much like the introduction- rather theoretical, and there is not sufficient discussion relating this study's findings to the rest of the literature. More importantly, it reads as a very selective report about just some of the results, only focusing on favorable outcomes. I realized that there is very little published human research about this supplement, but there still needs to be a more robust consideration of this study's findings, not just what might be happening mechanistically.
Author Response
Response to Reviewer 2 Comments
The title does not accurately reflect the study. The title is about a "cyclist" (singular, implying a case study), but the paper is about a group of "cyclists" (plural). Further, both groups (placebo and supplement) improved power in the Wingate test with no significant between group differences for this increase in power. The title is therefore misleading about the efficacy of the supplement vs placebo.
Response: We have changed "cyclist" to cyclists, as suggested. Regarding the title, we appreciate your comment; however, we want to clarify that we do not pretend to create any confusion or mislead the reader, since the improvements in performance level are given in the work zones (exercise at FTP and maximum power) during the incremental maximal test and not during the Wingate test.
The figures are very small and the text is hard to read. Can the authors please make these larger?
Response: As suggested, we have modified the figures to make them bigger.
Line 50: I suggest rewording this to say "exercise performance".
Response: As suggested, we have made the modification in the text. Line 51.
Line 54: The studies described in the following paragraph also describe supplementation effects on performance. Therefore, there is NOT only one study that has looked at "physical performance" effects of supplementation.
Response: There may have been a misunderstanding with what we wanted to convey in this paragraph. We have modified it (lines 55-56) to make clear that there was only one acute effects study in humans that examined 2S-hesperidin on performance. The rest of the studies that looked at physical performance in humans were chronic studies (lines 62). We hope that it is clearer now.
Line 87: missing space after "epicatechin"
Response: Amended. Line 353.
Lines 108-109: Are statements presented here also from reference 13?
Response: Yes, it refers to reference 13. The citation has been included. Line 112. Please note that this reference number has been updated to 14.
Line 112: How do you define "long-term" and "chronic"?
Response: We used long-term and chronic as synonyms to reflect non-acute situations. However, we understand that this may cause some confusion. Based on the suggestion by another reviewer, we have revised this sentence. Lines 113-117
Line 143: There should be a semi-colon instead of comma.
Response: Amended. Line 152.
Section 2.2: Please expand upon the randomization and placebo. How were participants assigned to their group? Was the placebo matched for visual appearance, taste, and smell?
Response: Information regarding randomization is now included. Lines 140-142. The capsules were similar in appearance. The taste and smell were hidden by the capsulated form. This information is now included. Lines 148-149.
Please explain how VO2R (body mass oxygen consumption) differs from VO2 (volume of oxygen uptake).
Response: We have included the definitions of the abbreviations to be clearer. Line 195-196. However, we are not sure that we have answered your comment. Please, could you be more specific? Thank you in advance.
Lines 192-193 Missing "a" before describing the tube (ex: a 3-mL tube vs 3 mL tube)
Response: Amended. Lines 209-210.
If the groups were not significantly different from each other in any parameters at baseline, why did you use baseline pre-test variables as covariates?
Response: For variables FatMax, VT1 and VT2, no baseline comparisons were performed. Thus, to avoid potential errors in the power generated at FatMax, VT1 and VT2, baseline values were used as covariates.
Please explain the purpose, validation, and use of the rectangular test in more depth.
Response: As suggested, we have included some data in the step test. The main feature of this test is that it has longer steps (10 min) than in the incremental test and the participants work with a defined intensity (W) in the incremental test. But in this section, we believe that there is no more information to provide. The aim of the step test was to find out the possible changes in oxygen and carbon dioxide consumption, fat and carbohydrate oxidation and efficiency using longer working steps than in the incremental test. This type of test is adequate for finding out metabolic changes, especially in submaximal zones of exercise. Lines 191-196.
Table 3 defines efficiency as "percentage". What percentage of what?
Response: In the step test methodology we have included the formula used for the calculation of energy efficiency. We have included a reference. Line 197.
Table 4 & Figure 4: I suggest removing the symbol for a trend for a significant change. The reader can see the p value and interpret that for him/herself.
Response: We appreciate your comment. However, we consider that the symbols are needed since the explanation of each of them and their corresponding p-value's range are presented on table footer.
Lines 293-294: Your conclusion is premature and not based on all of your data. There are plenty of other instances where there were no significant between-group effects (which is really what we're looking at when doing a placebo vs supplement trial). Please discuss the full picture of all of your data, not just significant changes that support your hypothesis.
Response: Thanks for your comment. We have removed the sentence "Thus, 2S-hesperidin does have a positive impact in the performance of amateur cyclists" to avoid confusions. Thus, this first paragraph of the discussion summarizes the main obtained outcomes. Lines 311-312.
Line 347: Should "is" be "in"?
Response: Amended. Line 377.
Line 348: I think to call that an "anti-aging effect" is overstepping the bounds of reasonable interpretation.
Response: We have introduced the term anti-aging from the article published by Biesemann et al., it is not our intention to increase any possible effect of hesperidin. However, we have deleted the sentence to not overstep the interpretation of their findings. Lines 378-380.
Line 353: "Justified" does not fit the context. Perhaps "demonstrated?
Response: As suggested, we have changed the term. Line 384.
Lines 363-364: Why the "y"?
Response: Amended. Line 393.
Line 392: Why "Sin embargo, Recently,"?
Response: Amended. Line 422.
Line 472: Missing a space between ROS & One
Response: Amended. Line 502.
The discussion reads much like the introduction- rather theoretical, and there is not sufficient discussion relating this study's findings to the rest of the literature. More importantly, it reads as a very selective report about just some of the results, only focusing on favorable outcomes. I realized that there is very little published human research about this supplement, but there still needs to be a more robust consideration of this study's findings, not just what might be happening mechanistically.
Response: Following your suggestion, the mechanistic part of the introduction has been removed and it has been kept in the discussion to explain some of the findings. As mentioned, studies in humans with hesperidin are scarce and this makes the discussion difficult, therefore, due to this lack of evidence in the scientific literature we have introduced the possible physiological mechanisms by which performance would be improved.

Round 2
Reviewer 2 Report
I appreciate the work that the authors have provided improving their manuscript. I read the responses both to my feedback and to the other reviewer, and I can value the changes made by the authors for strengthening this paper. Nonetheless, I do not think it is suitable for Nutrients. The conclusions and implications for sport seem overstated from the very small between group differences. The effect sizes are very small, reinforcing that changes were extremely small. I would also still like more detail about potential learning effects. What did the familiarization session for the Wingate test consist of? Did they actually do a Wingate in this familiarization time? I'm also concerned about the lack of correcting the alpha level for multiple comparisons. Thus, while this is an interesting study, I do not think the conclusions and implications match the story being told by the results.